# Structuring Benchmark into Knowledge Graphs to Assist Large Language Models in Retrieving and Designing Models

**Hanmo Liu[1,2]**    **Shimin Di[3,1]***   **Jialiang Wang[1]**    **Zhili Wang[1,2]**
**Jiachuan Wang[1]**    **Xiaofang Zhou[1]**    **Lei Chen[2,1]**
[1]HKUST    [2]HKUST (GZ)    [3]Southeast University
`{hliubm,sdiaa,jwangic,zwangeo,jwangey}@connect.ust.hk`
`{zxf,leichen}@ust.hk`

## Abstract

In recent years, the design and transfer of neural network models have been widely studied due to their exceptional performance and capabilities. However, the complex nature of datasets and the vast architecture space pose significant challenges for both manual and automated algorithms in creating high-performance models. Inspired by researchers who design, train, and document the performance of various models across different datasets, this paper introduces a novel schema that transforms the benchmark data into a Knowledge Benchmark Graph (KBG), which primarily stores the facts in the form of $performance(data, model)$. Constructing the KBG facilitates the structured storage of design knowledge, aiding subsequent model design and transfer. However, it is a non-trivial task to retrieve or design suitable neural networks based on the KBG, as real-world data are often off the records. To tackle this challenge, we propose transferring existing models stored in KBG by establishing correlations between unseen and previously seen datasets. Given that measuring dataset similarity is a complex and open-ended issue, we explore the potential for evaluating the correctness of the similarity function. Then, we further integrate the KBG with Large Language Models (LLMs), assisting LLMs to think and retrieve existing model knowledge in a manner akin to humans when designing or transferring models. We demonstrate our method specifically in the context of Graph Neural Network (GNN) architecture design, constructing a KBG (with 26,206 models, 211,669 performance records and 2,540,064 facts) and validating the effectiveness of leveraging the KBG to promote GNN architecture design.

## 1 Introduction

Designing neural networks (NNs) has traditionally been a complex and iterative process that relies heavily on human expertise and extensive experimentation (He et al., 2021). Manually crafting architectures requires a deep understanding of both the data and the underlying algorithms, posing significant challenges when dealing with diverse and intricate datasets. To alleviate this burden, automated methods have been developed to explore the vast architecture space more efficiently (Li & Talwalkar, 2019; Real et al., 2019; Zoph & Le, 2017). Recently, Large Language Models (LLMs) like GPT-4 have emerged as powerful tools that can assist in neural network design (Tornede et al., 2024). Leveraging their expansive knowledge and reasoning capabilities, LLMs can generate architectural suggestions and configurations for various types of neural networks, such as Convolutional Neural Networks (Zheng et al., 2023; Yu et al., 2023), Recurrent Neural Networks (Guo et al., 2024), and Graph Neural Networks (GNNs) (Wang et al., 2023a; Wei et al., 2023; Wang et al., 2024), offering new avenues for automating and enhancing the model design process.

Despite these advancements, designing NN architectures for unseen datasets remains time-consuming and demands iterative fine-tuning of configurations to achieve optimal performance on

---

*Corresponding author

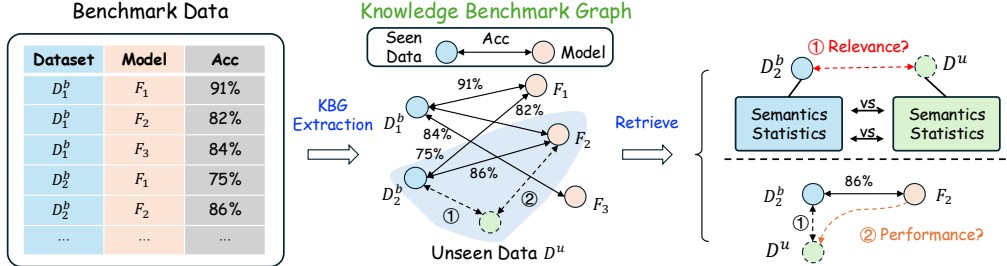

Figure 1: An illustration of constructing Knowledge Benchmark Graph and retrieving models.

specific tasks. This intricate procedure necessitates a profound understanding of both theoretical concepts and practical data considerations. Existing automated algorithms, such as traditional neural architecture search methods (He et al., 2021), aim to streamline this process but often do not leverage prior knowledge, frequently starting from scratch without considering existing design patterns or performance insights. Moreover, while LLMs have shown promise in suggesting NN architectures (Wang et al., 2023a; Dong et al., 2023), they tend to produce commonplace designs and may lack meticulousness in hyperparameter tuning (Wang et al., 2024). These limitations arise because they do not possess the detailed, structured knowledge necessary for customized model design tailored to specific datasets and tasks.

Recognizing these challenges, we observe that a wealth of design knowledge already exists in the form of benchmark data (Chitty-Venkata et al., 2023; Wang et al., 2024; Ying et al., 2019b), where scholars and experts have documented the performance of various models across different datasets. However, this valuable resource has not been effectively leveraged. To bridge this gap, we propose a novel schema that transforms benchmark data into a *Knowledge Benchmark Graph (KBG)* capable of storing the "data-model-performance" information. As shown in Fig. 1, entities in the KBG may include datasets and models, which can be further annotated with statistical properties, semantic descriptions, hyperparameter settings (see more details in Fig. 2). Then, the facts in KBG can capture the performance between data and models, including loss, accuracy, etc. This structured knowledge facilitates the retrieval and utilization of prior design insights, thereby aiding subsequent model design and transfer.

Intuitively, with a KBG that describes the "data-model-performance" relationship, automated algorithms only need to retrieve a well-performing model for a given dataset. However, utilizing this KBG to design an NN model is non-trivial. The KBG is constructed based on observed data and existing models, yet the data encountered by users are frequently unseen before (i.e., not included in the KBG). Consequently, establishing the relevance between datasets becomes a critical challenge that must be addressed. To tackle this challenge, we establish interconnections between datasets and models within the KBG. We evaluate the unsupervised relevance among datasets based on their statistical properties and semantic descriptions. This approach allows us to infer relationships and complete the KBG for datasets lacking direct performance records. However, it is worth noting that how to construct similarity between datasets is still an open and challenging topic. Not only should we consider the similarity of the data structure, domain, description, and statistics, but we also need to consider whether one model performs similarly on the two datasets. To aid future work on data similarity in model design, we propose a novel metric that prioritizes relevant insights effectively.

To validate that the above KBG idea can promote model design, we subsequently propose to integrate LLMs with the KBG to assist LLMs in designing the NN models. This integration enables automated algorithms to think and retrieve existing model knowledge in a manner akin to human experts when designing or transferring models. We demonstrate this process specifically in the context of Graph Neural Network architecture design, validating the potential of constructing and leveraging the benchmark KG to enhance LLM-driven network design. Our contributions are summarized as:

- We introduce a novel schema that transforms benchmark data into a Knowledge Benchmark Graph, summarizing the "data-model-performance" relationships. This KBG can also record information such as data statistics, model architecture, and model hyperparameters, which facilitate the structured storage of design knowledge to aid subsequent model design and transfer.

- To handle unseen data, we propose to model the relevance between datasets derived from inherent data characteristics. Furthermore, we also develop a novel evaluation metric to discuss how well the relevance scores prioritize useful insights.

- We validate the KBG idea in the context of GNN architecture design, where we construct a graph with 26,206 models, 211,669 performance records and 2,540,064 facts. Besides, we integrate the KBG with LLMs to promote the performance and efficiency of designing neural networks.

## 2 RELATED WORKS OF AUTOMATED MACHINE LEARNING

Automated Machine Learning (AutoML) has emerged as a critical approach to automate the design and optimization of machine learning models. Traditional AutoML methods aim to streamline the model development process by automatically processing data (Nargesian et al., 2017; Khurana et al., 2017), performing neural architecture search (NAS) (Ren et al., 2020), and tuning hyperparameters (Shahriari et al., 2016) without requiring extensive human expertise. The goal of neural network design is to identify the best-performing model $F^*$ for an unseen dataset $D^u$ from a large space $\mathbb{F}$:

$$\mathcal{M}(F^*; D^u) = \max_{F \in \mathbb{F}} \mathcal{M}(F; D^u), \tag{1}$$

where $\mathcal{M}(F; D^u)$ evaluates how a model $F$ performs on dataset $D^u$. This automation reduces the barriers for non-experts while enhancing the efficiency of model development for experts. Despite its potential, traditional AutoML faces significant challenges, particularly in its scalability and ability to generalize across a wide variety of datasets and tasks. On the one hand, the search space for neural architectures and hyperparameters is vast, making it computationally expensive to explore all possible configurations (typically several GPU hours or days) (Salehin et al., 2023; Liu et al., 2022), even accelerated by AutoML algorithms. On the other hand, the performance predictions of AutoML models are often limited to the datasets they are trained on, leading to suboptimal generalization to unseen data (Wen et al., 2019; Zheng et al., 2020).

Recently, to address the efficiency issue, there is a line of training-free AutoML methods (Mellor et al., 2020; Lopes et al., 2021) that estimates the suggested model performance by using a single forward or backward computation on a single minibatch of data without full-training (Tanaka et al., 2020; Xing et al., 2024). They design various score functions to evaluate the trainability and expressivity of the model, and prune the less promising models to reduce the search space (Xing et al., 2024). Although they share a similar object to our work, they still rely on the concrete training process to evaluate the model performance. Instead, we regard the "data-model-performance" information as a knowledge base to retrieve the suitable models without actual training.

**AutoML with Large Language Models.** Large Language Models (LLMs), such as GPT-4 (LLM, 2023) and LLaMA (LLM, 2024), have recently garnered significant attention for their capabilities, positioning them as powerful tools for automating various tasks (including model designs). Unlike traditional AutoML methods that rely heavily on search-based techniques, LLMs offer a more flexible and scalable alternative by leveraging their vast pre-trained knowledge to suggest model structures based on textual descriptions of the task at hand (Wei et al., 2023; Wang et al., 2023a; Dong et al., 2023; Wang et al., 2024).

In the context of AutoML, LLMs can interpret human-specified requirements through natural language prompts, enabling them to generate corresponding neural architectures tailored to specific tasks (Wang et al., 2024). For instance, users can guide LLMs to produce customized neural network configurations by providing detailed prompts that include dataset characteristics, desired performance metrics, and architectural preferences. Some approaches integrate LLMs as controllers within a NAS framework, where the LLM generates candidate architectures that are then evaluated and refined iteratively (Zheng et al., 2023; Zhang et al., 2023; Wang et al., 2023a). Specifically, by leveraging their few-shot learning capabilities (Brown et al., 2020), LLM controllers can utilize the optimization trajectory written in text to decide which model to validate next, incorporating additional domain-specific knowledge as a reference (Zhang et al., 2024; Nasir et al., 2024; Wang et al., 2024)—for example, recognizing that the GraphSAGE (Hamilton et al., 2017a) model is better suited for handling dense graphs. Additionally, LLMs can assist in generating code for model implementation, further streamlining the development process (Cheng et al., 2023).

Despite these advancements, challenges remain in fully leveraging LLMs for AutoML tasks. LLMs may lack detailed domain-specific knowledge required for tuning hyperparameters or may generate architectures that are common but not necessarily optimal for a given dataset (Wang et al., 2024). Integrating LLMs with structured knowledge sources, such as KGs or deep learning benchmarks, can enhance their capability to produce more customized and effective neural network designs.

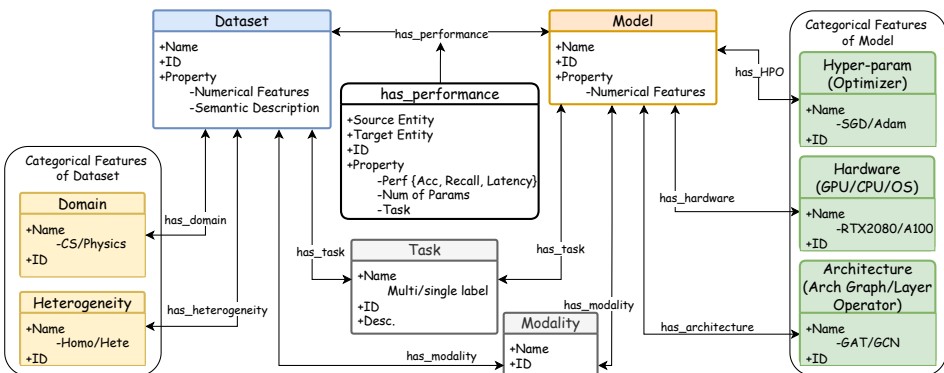

Figure 2: An architectural diagram of the Knowledge Benchmark Graph.

## 3 PROBLEM DEFINITION

In equation 1, it is quite time-consuming to obtain the accurate performance $\{\mathcal{M}(F; D^u)\}$ of many candidate models $F \in \mathbb{F}$ on an unseen dataset $D^u$. To efficiently obtain $\mathcal{M}(F; D^u)$ when searhcing for $F^*$, some classic methods have been proposed in the early years. For example, Domhan et al. (2015) propose to explore the learning curves to predict the performance of a model on a dataset. Furthermore, one-shot NAS proposes the weight sharing mechanism to avoid the computational overhead of training many candidate models (Pham et al., 2018). However, these methods still require a non-negligible computational overhead, since they all need to train models somehow. Inspired by many well-established benchmarks Chitty-Venkata et al. (2023), we may recommend those benchmark models $F^b \in \mathbb{F}^b$ that are effective for benchmark data sets $D^b \in \mathbb{D}^b$ to unseen data $D^u$. In such a way, the complex model design problem in equation 1 can be simplified as follows.

**Definition 1 (The Problem of Retrieving Models from Benchmark to Unseen Data)** *Given an unseen dataset $D^u$ and a model search space $\mathbb{F}^b$ with performance records on benchmark datasets $\mathbb{D}^b$, the goal is to find an effective model $F^b$ for $D^u$ by measuring the similarity between benchmark $\mathbb{D}^b$ with unseen $D^u$, and retrieving most effective models from $\mathbb{F}^b$:*

$$\max_{F \in \mathbb{F}} \mathcal{M}(F; D^u) \propto \max_{F \in \mathbb{F}^b} \mathcal{M}(F; D^b) \cdot \mathcal{S}(D^b, D^u), \tag{2}$$

*where $\mathcal{S}(D^b, D^u)$ evaluates the similarity between benchmark data set $D^b$ with unseen data $D^u$.*

There are two main intuitions behind Def. 1. First, if the given data is one we have seen before, we only need to find the one model with the outstanding performance from a well-established benchmark record (see Sec. 4). Secondly, as we only need to design models for unseen data $D^u$, we can find the benchmark dataset $D^b \in \mathbb{D}^b$ that is similar to $D^u$, and then transfer the most effective models from $\mathbb{F}^b$ on $D^b$ to $D^u$. This approach aligns with the intuitive principle: "*Similar datasets prefer similar models*" (Bardenet et al., 2013). It is worth noting that transfer learning (Pan & Yang, 2009) focuses more on how to reuse a specific model in a source domain on a target domain to alleviate the data scarce issue. Def. 1 focuses more on how to quickly recommend a model $F^b$ from massive benchmark models $\mathbb{F}^b$ and data sets $\mathbb{D}^b$ that can achieve high performance on unseen data $D^u$.

To leverage benchmark data, two challenges need to be addressed: 1) **Organizing Benchmark Data**: The organization of benchmark data largely affects storage efficiency and information retrieval effectiveness. We address this challenge by proposing the knowledge benchmark graph (KBG) in Sec. 4. 2) **Retrieve Effective Models**: As the benchmark knowledge does not directly translate to unseen datasets, developing a robust $\mathcal{S}(\cdot)$ that prioritizes the most relevant $D^b$ for $D^u$ and selecting the most suitable models to $D^u$ based on the identified similar datasets are non-trivial tasks. We propose the retrieval method in Sec. 5 to address this challenge. Since $\mathcal{S}(\cdot)$ is the foundation of this novel retrieval process, we further propose a comprehensive evaluation metric in Sec. 6.

## 4 BENCHMARK KNOWLEDGE GRAPH CONSTRUCTION

Benchmark data, traditionally stored in tabular formats (Ying et al., 2019a; Dong & Yang, 2020; Qin et al., 2022), contains valuable information about model performance across various datasets.

Using traditional tabular formats to represent diverse model structures across multiple datasets poses challenges, such as missing values and inefficient storage, which complicates data management and retrieval. To better utilize the benchmark data, we transform it into a Knowledge Benchmark Graph (see Fig. 2). The KBG can organizes datasets, models, and performance as entities and relations, enabling flexible, context-aware queries based on relationships between entities rather than simple row-column searches. This structure improves data accessibility and interpretability, allowing LLMs to process and respond to queries more effectively. We next introduce the KBG in detail.

**Entity and Its Attributes.** To record the "data-model-performance" information, the KBG mainly has following entities:

1. **Data:** As a fundamental component, `Data:{data_ID, data_Name, data_Property}` records information about the datasets.
   - `data_Property:{statistics:{⋯}, description:{⋯}}` are the attributes that provide high-level information about the dataset, including statistical features and a semantic description. Detailed feature information and examples are provided in the Appx.A.3.

2. **Model:** A neural network model generally consists of three main parts: model architecture, hyperparameters, and software/hardware configurations. To reduce redundancy, KBG uses `Model:{model_ID, model_Name, model_Property}` to store models and model-specific attributes (typically numerical, such as dropout ratio), while common attributes are stored as other entity types (see **Other Entity Types** for more details).
   - `model_Property:{numerical_Features:{⋯}}` are the attributes that include the numerical configurations of the model, such as learning rates, epochs, and hidden units.

3. **Other Entity Types**: These entities store categorical features of datasets and models, using the format `entity_Type:{entity_ID, entity_Name}` with actual entity names introduced below. Generally, they can be divided into three groups.
   - Common Attributes: `task` and `modality` are essential for determining the relatedness of datasets and models and for selecting suitable ones.
   - Data-related Attributes: Semantic aspects of the datasets, such as their `domain` and `heterogeneity` (for graph datasets), are represented as entities.
   - Model-related Attributes: Important categorical features for models include software/hardware information like `operating_System`, `CPU`, and `GPU`, model architecture components like `architecture_Graph` and `layer_Operator`, and hyperparameter configurations like `optimizer`.

Because the categorical features are widely shared across datasets and/or models, transforming them into entities reduces the complexity in the storage and allows a higher extendibility for new categories. On the other hands, the numerical features and semantic descriptions are often unique to the corresponding entities, thus keeping them in the attributes is more applicable.

**Relations.** There are two types of relations in the KBG:

1. **Performance**: KBG utilize the hyper-relational facts (Rosso et al., 2020) $r(e_h, e_t, \{r^o : v^o\}_{o=1}^{N_o})$, to store the performance of different models on various datasets, such as `hasPerformance(data_ID, model_ID, {Accuracy:⋯, Loss:⋯, ⋯})`. Generally, `hasPerformance(data_ID, model_ID)` is regarded as main triplet, while $\{\texttt{Accuracy}:⋯, \texttt{Loss}:⋯\}$ are role-value pairs $\{r^o : v^o\}_{o=1}^{N_o}$.

2. **Feature**: The relations between the datasets/models and their kinds of properties are stored as binary facts `hasFeature:{data_ID/model_ID, feature_ID}`, where the actual relation names vary with the corresponding features.
   - As shown in Fig. 2, such relations include `has_domain`, `has_task`, `has_modality`, `has_architecture`, `has_hardware`, and `has_HPO`, etc.

## 5 RETRIEVING MODELS WITH KBG

Based on the KBG constructed in Sec. 4, it is now important to retrieve the similar datasets and and relevant models for unseen data. Thus, we design dataset similarity score $\mathcal{S}(\cdot, \cdot)$ and model relevance score $\mathcal{R}(\cdot, \cdot)$ for retrieval correspondingly.

**Data Similarity.** Following Def. 1, designing an effective dataset similarity score $\mathcal{S}(\cdot)$ is the crucial part. As a first approach, we directly use the statistical features of datasets to measure the similarity.

Let the feature vectors of the benchmark dataset $D^b$ and the unseen dataset $D^u$ be represented as $d$-dimensional vectors $\mathbf{v}^b$ and $\mathbf{v}^u$, respectively. The dataset similarity score $\mathcal{S}(\mathbf{v}^b, \mathbf{v}^u)$ is defined as:

$$\mathcal{S}(\mathbf{v}^b, \mathbf{v}^u) := 1 - \frac{1}{d} \sum_{d' \leq d} \min \left\{ (\mathbf{v}^b_{d'} - \mathbf{v}^u_{d'})^p, \alpha \right\},$$

where $\alpha$ represents the maximum allowable difference for any feature dimension, and $p$ is a power parameter that controls the sensitivity to feature differences. Both $\alpha$ and $p$ are hyperparameters. The similarity is ranged within $[0, 1]$ with a higher value for a more similar dataset. Empirically, we set $p = \frac{1}{3}$ and $\alpha = 0.8$ to mitigate the impact of outlier feature differences.

**Model Relevance.** While $\mathcal{S}(\cdot, \cdot)$ helps prune the model search space, directly transforming models from the most similar $D^b$ to $D^u$ can not guarantee a good performance. Moreover, a less similar dataset may also assist defining the model performance on $D^u$. Thus, to better decide the relevance of model $F$ to $D^u$, it is necessary to aggregate the model performances on multiple benchmark datasets that are similar to $D^u$. The model relevance score $\mathcal{R}(\cdot, \cdot)$ is defined as:

$$\mathcal{R}(F; D^u) = \sum_{D^b \in \mathbb{D}^c} \mathcal{M}(F; D^b) \cdot \mathcal{S}(D^b, D^u), \tag{3}$$

where $\mathbb{D}^c = \{D^b | \mathcal{S}(D^b, D^u) \geq \delta, D^b \in \mathbb{D}^b\}$. $\mathcal{R}(\cdot, \cdot)$ is a reflection of Def. 1 which captures both the model's historical performance and the similarity of the benchmark dataset to the unseen dataset.

Please note that this paper focuses more on how to avoid training as much as possible by leveraging the prior information of KBG itself. So that we only propose relatively simple $\mathcal{S}(D^b, D^u)$ and $\mathcal{R}(F; D^u)$ to demonstrate the feasibility of the proposed method. The more advanced and accurate designs of the scores can be easily combined with ours with a computational overhead.

# 6 EVALUATING DATA SIMILARITY

As discussed in Sec. 3, the quality of data similarity metric $\mathcal{S}(D^b, D^u)$ largely affects the model retrieval performance. However, given the novelty of the problem, there lacks a comprehensive evaluation metric to assess the effectiveness of $\mathcal{S}(D^b, D^u)$. While the performance on $D^u$ is an intuitive measurement, it does not directly indicate whether the model transfer is successful. Thus, we first introduce factors that directly reflects the transfer performance of retrieved models in Sec. 6.1. Then based on these factors, we propose a novel evaluation metric for $\mathcal{S}(D^b, D^u)$ in Sec. 6.2.

## 6.1 FACTORS OF DATA SIMILARITY EFFECTIVENESS

When retrieving models from the KBG for $D^u$, sufficiently similar $D^b$ datasets should yield models that perform well on $D^u$. Thus, the effectiveness of $\mathcal{S}(D^b, D^u)$ is measured by how well top-performing models from $D^b$ transfer to $D^u$. Formally, the problem can be defined as follows:

**Definition 2 (The Problem of Transferring Models in KBG to Unseen Data)** *Given an unseen dataset $D^u$ and a model search space $\mathbb{F}^b$ with performance records on benchmark datasets $\mathbb{D}^b$ stored in KBG, the goal is to design a dataset similarity metric $\mathcal{S}(\cdot)$ such that, if $\mathcal{S}(D^b, D^u) \geq \delta$ and a subset of models satisfies $\mathbb{F}^c = \{F \in \mathbb{F}^b \mid \mathcal{M}(F; D^b) \geq \tau\}$, then the probability that $\mathcal{M}(F; D^u)$ is close to $\mathcal{M}(F; D^b)$ (within an error tolerance $\epsilon$) is at least $\Delta$:*

$$P\left(\mathcal{M}(F; D^u) \geq \mathcal{M}(F; D^b) - \epsilon \mid F \in \mathbb{F}^c\right) \geq \Delta, \tag{4}$$

*where $\delta$ and $\tau$ are the thresholds for similarity and performance, $\epsilon$ is the error tolerance, and $\Delta$ is the probability lower-bound.*

Following Def. 2, $\mathcal{S}(\cdot, \cdot)$ is evaluated by two key factors, $\epsilon$ and $\Delta$. The value $\epsilon$ measures how well transferred models maintain high performance, while $\Delta$ represents the probability of successful transfer given $\epsilon$. Intuitively, a smaller error tolerance $\epsilon$ lowers the success probability $\Delta$, and an effective $\mathcal{S}(\cdot, \cdot)$ should to minimize $\epsilon$ while maximizing $\Delta$.

We further study the relationships between $\epsilon$ and $\Delta$ with $\mathcal{S}(\cdot, \cdot)$ defined in equation 3 and the KBG constructed from NAS-Bench-Graph (Qin et al., 2022). We select one dataset from KBG as $D^u$ and treat the most similar remaining dataset as $D^b$. In order to compare performances across different datasets, we define $\mathcal{M}(\cdot)$ as a relative measure of performance, comparing a model $F$ to the theoretically optimal model $F^*$ in $\mathbb{F}^b$, $\mathcal{M}(F; D^b) := \frac{\mathcal{M}_{abs}(F; D^b)}{\mathcal{M}_{abs}(F^*; D^b)}$. We transfer models with performance no less than $\tau = 0.990$ on $D^b$ to $D^u$ and compute the transferred performance error $\epsilon$ and confidence $\Delta$. As shown in Fig. 3, $\Delta$ increases monotonically with $\epsilon$, and when $\epsilon$ is within 0.04, $\Delta$ is at least 20% for all datasets. This also confirms the feasibility of transferring high-performing models to similar datasets.

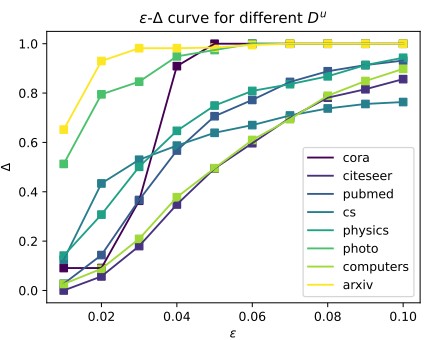

Figure 3: The probability that the transferred model's performance is within $\epsilon$.

**Remark 6.1** *How to design a good similarity metric $\mathcal{S}(D^b, D^u)$ is an open question. The core idea of Def. 2 is that a better similarity metric should lead to smaller $\epsilon$ but larger $\Delta$ in equation 4. The Def. 2 serves as an guidance for subsequent work to evaluate different $\mathcal{S}(D^b, D^u)$.*

## 6.2 DATA SIMILARITY EVALUATION

Following Def. 2, retrieved models $\mathbb{F}^c$ from $D^b$ to $D^u$ have a better transferability if they have a higher possibility $\Delta$ of being transferred with a low performance drop $\epsilon$. This is equivalent to lifting the $\epsilon$-$\Delta$ curve to the upper left corner and results in a large area under the curve, as illustrated in Fig. 3. Thus, we can quantify the transferability $\mathcal{T}(\cdot)$ of $\mathbb{F}^c$ by measuring the area under the $\epsilon$-$\Delta$ curve, i.e. $\mathcal{T}(\mathbb{F}^c) := \int_0^{\epsilon_{max}} \Delta(\epsilon)d\epsilon$. $\epsilon_{max}$ is a hyper-parameter that limits the largest transferred error, as a too large error is meaningless. Considering that the exact relationship between $\epsilon$ and $\Delta$ is unknown and needs to be empirically evaluated, we use the estimated version of $\mathcal{T}(\cdot)$ in practice:

$$\hat{\mathcal{T}}(\mathbb{F}^c) = \frac{1}{N} \sum_0^{\epsilon_{max}} \Delta(\epsilon), \tag{5}$$

where $N$ is the number of estimated $\epsilon$ values. A more transferable $\mathbb{F}^c$ has a larger score close to 1, showing the performances of $\mathbb{F}^c$ sustain across two datasets. This score serves as the ground truth similarity between $D^b$ and $D^u$, and a good $\mathcal{S}(D^b, D^u)$ should align with this observation.

Then, given a set of benchmark datasets $\{D_i^b\}_i$ and a set of candidate models $\{\mathbb{F}_i^c\}_i$ separately selected from each benchmark dataset with the same standard, we evaluate $\mathcal{S}(\cdot, \cdot)$ by the linearity between the similarity and the transferability. Following the definition of $R^2$ score, we define the *Relevance Linearity Score* (RLS) as:

$$\text{RLS}(\mathcal{S}) = 1 - \frac{\sum_{D_i^b \in \{D_i^b\}_i}(\mathcal{S}(D_i^b, D^u) - \hat{\mathcal{T}}(\mathbb{F}_i^c))^2}{\sum_{D_i^b \in \{D_i^b\}_i}(\mathcal{S}(D_i^b, D^u) - \hat{\mathcal{T}}_{avg})^2},$$

where $\hat{\mathcal{T}}_{avg} = 1/|\{D_i^b\}_i| \cdot \sum_{D_i^b \in \{D_i^b\}_i}(\mathcal{T}(\mathbb{F}_i^c))$ is the average transferability score of $\{\mathbb{F}_i^c\}_i$. A higher RLS indicates a better linearity between the similarity and the transferability, suggesting that $\mathcal{S}(\cdot, \cdot)$ correctly ranks the dataset similarity by the model performances on the unseen datasets.

## 7 EXPERIMENT

### 7.1 EXPERIMENT SETUP

**Implementation Details.** To verify the effectiveness of KBG and integration of LLMs with KBG, the benchmark datasets used in our experiments is the NAS-Bench-Graph dataset (Qin et al., 2022),

Table 1: The statistics of KBG

| # Models | # Datasets | # Perf. Rec. | # Entities | # Facts |
|---|---|---|---|---|
| 26,206 | 9 | 211,669 | 211,712 | 2,540,064 |

Table 2: The initial performance comparison (absolute accuracy) of GNN designed by different works. We mark the best performance in **bold**, and and second best in underline.

| Type | Model | Cora | Citeseer | PubMed | CS | Phys. | Photo | Comp. | arXiv |
|------|-------|------|----------|--------|-----|-------|-------|-------|-------|
| Manual | GCN | 80.97 | 69.90 | **77.46** | 88.65 | 90.85 | 89.44 | **83.16** | 71.08 |
| | GAT | 80.83 | **70.70** | 75.93 | 88.72 | 89.47 | 89.93 | 81.35 | 71.24 |
| | SAGE | 79.47 | 66.13 | 75.50 | 87.81 | 91.43 | 88.29 | 81.46 | 70.78 |
| | GIN | 79.77 | 63.30 | 76.74 | 81.08 | 86.67 | 87.37 | 73.95 | 61.33 |
| | ChebNet | 79.40 | 67.03 | 75.13 | 89.50 | 89.75 | 86.65 | 79.10 | 70.87 |
| | ARMA | 78.33 | 66.20 | 75.00 | 89.87 | 88.88 | 86.55 | 78.47 | 70.87 |
| | k-GNN | 78.06 | 30.97 | 75.38 | 83.81 | 88.98 | 86.45 | 76.31 | 63.18 |
| Classic | Random | 77.87 | 66.64 | 74.16 | 81.78 | 90.59 | 89.04 | 76.61 | 68.93 |
| | EA | 78.23 | 66.40 | 72.88 | 87.03 | 88.07 | 87.30 | 77.56 | 68.28 |
| | RL | 73.44 | 65.35 | 75.44 | 86.17 | 88.15 | 89.48 | 77.70 | 68.00 |
| AutoML | GNAS | 78.55 | 63.25 | 73.04 | 86.04 | 89.54 | 87.27 | 70.96 | 69.94 |
| | Auto-GNN | 78.58 | 65.60 | 76.07 | 89.06 | 89.26 | 89.34 | 77.49 | 70.62 |
| | GPT4GNAS | 78.50 | 67.46 | 73.89 | 89.26 | 89.44 | 89.12 | 77.21 | 68.98 |
| | GHGNAS | 79.13 | 67.35 | 74.90 | 89.15 | 88.94 | 89.42 | 77.04 | 69.66 |
| | DesiGNN-init | 80.31 | 69.20 | 76.60 | **89.64** | **92.10** | **91.19** | 82.20 | 71.50 |
| Sim. | Kendall | 67.73 | 69.20 | 71.80 | 88.56 | 91.56 | 88.90 | 76.85 | 71.49 |
| | Overlap | 79.36 | 67.30 | 71.80 | 88.56 | 89.95 | 90.37 | 76.85 | **71.68** |
| | Ours | **82.53** | 69.20 | 76.53 | 89.32 | 90.34 | 90.37 | 76.61 | **71.68** |

which includes performance records for 26,206 GNNs (Zhou et al., 2020) evaluated across 9 different graph datasets. The search space consists of GNN model architectures, represented as directed acyclic graphs (DAGs) with 4 nodes and 9 possible layer types. The 9 datasets used in the evaluation are: Cora, Citeseer, Pubmed, CS, Physics, Photo, Computers, Arxiv, and Proteins. The detailed statistics are provided in Tab.8 and Appx. A.3. [1]

**Evaluation Metric.** The performance of a suggested model design $F \in \mathbb{F}$ on an unseen dataset $D^u$ is evaluated as its accuracy $\mathcal{M}(F; D^u)$. For retrieval methods, $F$ is directly retrieved from KBG based on the performance records of similar datasets $\mathbb{D}^b \backslash D^u$, i.e., removing the records to related to $D^u$ from KBG. While for AutoML methods, $F$ is generated by the their algorithms.

**Baselines.** We compare our approach against a diverse range of baselines:

- **Manually Designed GNNs**: We include several popular GNNs commonly used in the literature, such as GCN (Kipf & Welling, 2017), GAT (Velickovic et al., 2017), GraphSAGE (Hamilton et al., 2017b), GIN (Xu et al., 2019), ChebNet (Defferrard et al., 2016), ARMA (Bianchi et al., 2019), and k-GNN (Morris et al., 2019).

- **Classical NAS Algorithms**: We compare against Random Search (Li & Talwalkar, 2019), Evolutionary Algorithms (Real et al., 2019), and Reinforcement Learning (Zoph & Le, 2017).

- **AutoML Methods for GNNs**: We include state-of-the-art AutoML methods specifically designed for GNN architecture search, such as GNAS and Auto-GNN (Gao et al., 2020; Zhou et al., 2022).

- **LLM-Based AutoML Methods**: Given the recent success of large language models (LLMs) in AutoML, we include GPT4GNAS (Wang et al., 2023a), GHGNAS (Dong et al., 2023), and DesiGNN (Wang et al., 2024) as LLM-based AutoML baselines.

- **Similarity Metrics**: As we retrieve model designs based on data similarity in Sec. 5, we compare against other similarity metrics such as Kendall (You et al., 2020) and Overlap (Wang et al., 2024), which measure similarity based on performance differences of one model across different datasets.

### 7.2 EFFECTIVENESS OF RETRIEVING MODEL DESIGNS WITH KBG

The key to applying the KBG is whether an effective model from a similar dataset is still effective on the unseen dataset. To validate this, we simply retrive the best performing model from the most similar benchmark datasets to the unseen data, and report its performance on the unseen data. As shown in Tab. 2, our naive retrieval method achieves the best or second-best performance on 4 out of 8 datasets, and is competitive on the remaining 4 datasets. Given that other methods require repeated training, which is time-intensive, this result demonstrates the effectiveness of using a KBG to assist model design. Additionally, compared to Kendall and Overlap, which rely on supervised model

---

[1]The code is available at `https://github.com/liuhanmo321/kgnas`.

Table 3: The RLS of different similarity metrics in initial model suggestion.

| Metric | Cora | Citeseer | PubMed | CS | Phys. | Photo | Comp. | Arxiv |
|--------|------|----------|--------|-----|-------|-------|-------|-------|
| L1 | 0.452 | 0.126 | 0.003 | 0.020 | 0.013 | 0.310 | 0.025 | 0.014 |
| L2 | **0.498** | 0.064 | 0.012 | 0.007 | **0.040** | **0.328** | 0.036 | 0.012 |
| Ours | 0.435 | **0.340** | **0.004** | 0.025 | 0.002 | 0.252 | **0.043** | **0.027** |

Figure 5: $\epsilon$-$\Delta$ curve when considering Cora, Physics, and Arxiv as unseen datasets. The legends denote the benchmark datasets and their similarites to the unseen datasets. $\tau$ is fixed to 0.980.

performance on unseen data, our metric achieves better performance with only intrinsic dataset characteristics. This highlights the effectiveness of leveraging dataset characteristics for model transfer.

### 7.3 THE STUDY ON THE IMPACT OF $\tau$, $\epsilon$, AND $\Delta$

With the dataset similarity metric validated as effective for model design, we further investigate the definition of model transferability based on Def.2. Recall that an effective candidate model set $\mathbb{F}^c$ ($\geq \tau$) should have a high success probability ($\Delta$) with a low error threshold ($\epsilon$). As

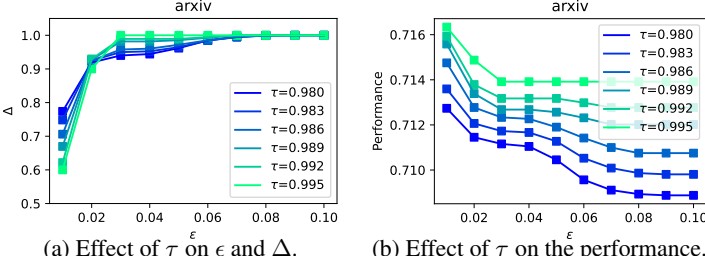

(a) Effect of $\tau$ on $\epsilon$ and $\Delta$.   (b) Effect of $\tau$ on the performance.

Figure 4: Effect of performance threshold $\tau$ on $\epsilon$, $\Delta$ and performance.

shown in Fig. 4, we study the impact of increasing $\tau$ on $\epsilon$ and $\Delta$. Fig. 4a illustrates that maintaining a high $\Delta$ at a low $\epsilon$ is challenging, a trend consistent across all values of $\tau$. However, when comparing Fig. 4a to Fig. 4b, we observe that the transferred models with a smaller $\epsilon$ generally exhibit better performances. This underscores the importance pursuing a small $\epsilon$ for successful model transferability. Additionally, Fig. 4b supports our assumption that a better-performing model (larger $\tau$) from a benchmark dataset is more likely to perform well on an unseen dataset. The figures on other datasets are presented in the Appx.A.6.

### 7.4 THE STUDY ON THE SIMILARITY SESIGN WITH EVALUATION METRIC

As shown in Fig. 5, we study how the similarity between $D^u$ and $D^b$ affects the model transfer performance. Intuitively, a more similar $D^b$ should have a higher possibility $\Delta$ of transfering models within a low performance drop $\epsilon$, which is reflected in a larger area under the $\epsilon$-$\Delta$ curve. The results in Fig. 5 show that the current similarity metric provides a a generally accurate ranking for the top three similar datasets. However, it still fails to rank the truly most similar dataset as the highest.

Following RLS defined in Sec. 6.2, we next present scatter plots between transferability score $\hat{\mathcal{T}}(\mathbb{F}^c)$, i.e. the $\epsilon$-$\Delta$ curve area, and dataset similarity in Fig. 6. We expect the curve area to be linear to the similarity, but the results indicate that the current similarity metric is flawed in achieving this goal. However, when comparing the RLS ($R^2$ score) with the model performance in Tab. 2, we observe that when the metric has a higher RLS on an unseen dataset, the retrieved similar dataset tends to have more transferrable model design. For example, our $\mathcal{S}(\cdot, \cdot)$ has a higher RLS on Cora, and the retrieved model has the best performance as well. This validates the effectiveness of using linearity as an evaluation metric for dataset similarity. Furthermore, we evaluate the different designs of $\mathcal{S}(\cdot, \cdot)$ with RLS in Tab. 3. It can be seen that no single metric is effective for all datasets. More detailed comparisons and analysis are provided in Appx. A.7.

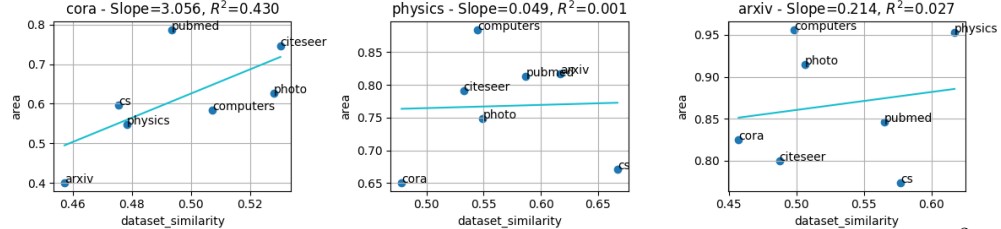

Figure 6: The linearity between the dataset similarity and the area below $\epsilon$-$\Delta$ curve. The $R^2$ score denotes the RLS metric.

### 7.5 IMPACT OF USING DIFFERENT SIZES OF $\mathbb{D}^c$ ON MODEL SUGGESTION

The previous experiments focused on recommending models using a single similar dataset. However, the KBG structure allows recommendations from multiple similar datasets. In Tab. 4, we analyze how incorporating different datasets affects model performance with $\mathcal{R}(\cdot)$ defined in equation 3. When more similar datasets are considered, the recommended models can outperform those from initially less optimal datasets. However, the improvement is inconsistent, likely due to the current simplistic combination method. With a more sophisticated approach to measure data similarity, leveraging multiple similar datasets is expected to result in better model recommendations.

Table 4: The effectiveness of model relevance score $\mathcal{R}(\cdot)$ with different $|\mathbb{D}^c|$. $|\mathbb{D}^c|$=1 is the same as the naive recommendation. We mark the best performance in **bold**, and and second best in underline.

| Method | Cora | Citeseer | PubMed | CS | Phys. | Photo | Comp. | arXiv |
|---|---|---|---|---|---|---|---|---|
| DesiGNN-init | 80.31 | **69.20** | **76.60** | **89.64** | 92.10 | 91.19 | 82.20 | 71.50 |
| $\mathcal{R}(\cdot)$, $|\mathbb{D}^c|$=1 | **82.53** | **69.20** | 76.53 | 89.32 | 90.34 | 90.37 | 76.61 | **71.68** |
| $\mathcal{R}(\cdot)$, $|\mathbb{D}^c|$=2 | 79.16 | 68.40 | 75.63 | 89.57 | 92.05 | 91.19 | **83.39** | 70.33 |
| $\mathcal{R}(\cdot)$, $|\mathbb{D}^c|$=3 | 79.63 | 68.73 | 76.46 | 89.45 | **92.11** | **91.48** | 81.02 | 71.09 |

### 7.6 INCOOPERATION WITH LARGE LANGUAGE MODEL

As an important application scenario of KBG, we present the study on the effectiveness of incoopering LLMs with KBG to suggest models in Tab. 5. Given an unseen dataset, we first retrieve candidate models $\mathbb{F}^c$ from KBG with $\mathcal{R}(\cdot)$ in equation 3. Then we require the LLM to infer or select a model $F'$ from $\mathbb{F}^c$. Afterwards, we repeatedly retrieve another set of candidates whose architectures and performances across $\mathbb{D}^c$ are both similar to $F'$ and ask LLM to refine $F'$. Results show that combining KBG with LLM outperforms the initial models from DesiGNN and only KBG. Besides, it can be seen that selecting a model design with LLMs is more effective than inferring one. This shows that retrieved models are already effective, and forcing LLMs to infer a new model design may not be necessary. Experiment details are provided in Appx.A.8.

Table 5: The performance of incooperating LLMs with KBG to suggest models.

| Type | Method | Cora | Citeseer | PubMed | CS | Phys. | Photo | Comp. | arXiv |
|---|---|---|---|---|---|---|---|---|---|
| Initial | DesiGNN-Init | 80.31 | 69.20 | 76.60 | **89.64** | 92.10 | 91.19 | 82.20 | 71.50 |
| | KBG-Init | **82.53** | 69.20 | 76.53 | 89.32 | 90.34 | 90.37 | 76.61 | 71.68 |
| Refined | KBG+LLM Infer | 80.32 | **69.56** | 76.60 | 89.53 | **92.61** | 91.87 | 81.26 | 71.69 |
| | KBG+LLM Select | 80.54 | 69.06 | **77.23** | 89.53 | **92.61** | **91.95** | **82.34** | **72.05** |

## 8 CONCLUSION

In conclusion, the Knowledge Benchmark Graph (KBG) presents a novel approach to enhance neural network design by structuring benchmark data into a graph that stores model performance across various datasets. The KBG facilitates efficient and automated model designs for unseen datasets by retrieving insightful knowledge from prior benchmarks. However, this paper still leave several challenges unsolved. For example, we may construct more accurate similarity functions between seen dataset and unseen one. Besides, it is meaningful to combine existing benchmarks together to build a larger KBG to support model design in different domains. In the subsequent research, we will aim to combine KBG with automatic learning systems and take a step towards a smarter and more efficient neural network design process.

ACKNOWLEDGMENTS

Lei Chen's work is partially supported by National Key Research and Development Program of China Grant No. 2023YFF0725100, National Science Foundation of China (NSFC) under Grant No. U22B2060, Guangdong-Hong Kong Technology Innovation Joint Funding Scheme Project No. 2024A0505040012, the Hong Kong RGC GRF Project 16213620, RIF Project R6020-19, AOE Project AoE/E-603/18, Theme-based project TRS T41-603/20R, CRF Project C2004-21G, Guangdong Province Science and Technology Plan Project 2023A0505030011, Guangzhou municipality big data intelligence key lab, 2023A03J0012, Hong Kong ITC ITF grants MHX/078/21 and PRP/004/22FX, Zhujiang scholar program 2021JC02X170, Microsoft Research Asia Collaborative Research Grant, HKUST-Webank joint research lab and 2023 HKUST Shenzhen-Hong Kong Collaborative Innovation Institute Green Sustainability Special Fund, from Shui On Xintiandi and the InnoSpace GBA.

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

# A   APPENDIX

## A.1   KNOWLEDGE GRAPH DEFINITION

A knowledge graph (KG), denoted as $G = \{E, R, H\}$, is a structured representation of knowledge, where $E$ represents the set of entities, $R$ the set of relations, and $H$ the set of facts. Each fact $h \in H$ is expressed as a triple $(e_1, r, e_2)$, where $e_1, e_2 \in E$ are entities, and $r \in R$ is a relation that connects them. These facts illustrate how entities are linked by specific relations, forming a graph structure that can be queried for relevant information. To incorporate more detailed information, entities and relations can have their own attributes, denoted as $Q$, where $Q = \{(q_i, v_i)\}$ represents a set of key-value pairs, with $q_i$ as the attribute type (key) and $v_i$ as the corresponding value. When relations are enriched with attributes, they are referred to as hyper-relations and their facts become quadruples $(e_1, r, e_2, Q)$.

## A.2   ADDITIONAL RELATED WORKS

Another line of research focuses on leveraging existing dataset-model relationships to recommend models for unseen datasets (Wu et al., 2020; Tan et al., 2024). Different from our approach aimed at designing new models, these methods prioritize the application of pre-trained models, which are combined with their specifications to form learnware—a structured knowledge representation for future model recommendations. To enable effective matching between learnware and unseen datasets, (Wu et al., 2020) employs a reduced kernel mean embedding method. More recently, Beimingwu (Tan et al., 2024) has integrated the learnware application pipeline into a comprehensive platform, encompassing preparation, matching, and deployment processes.

## A.3   KNOWLEDGE BENCHMARK GRAPH INFORMATION

Here we provide the examples about the structures of the Knowledge Benchmark Graph (KBG). We use the Nas-Bench-Graph, which is a benchmark dataset with performance records on 26,206 models for 9 graph datasets, to construct the KBG. The important entities are `Data` and `Model`, as they are the have attributes for their detailed informaiton.

- `Data`: as claimed in the main text, the `Data` entity represents the dataset used for training and evaluating the models. As we are using the graph datasets, the numerical features are extracted from the graph structure by using the NetworkX library `https://networkx.org/`. In case the graph is too large to be processed, we randomly sample subgraphs of the original graph to calculate its statistics and note the corresponding attributes with 'local'. Fot other data modalities, their numerical features can be replaced by the corresponding statistical characteristics. The attributes of the `Data` entity include:
  - `node_feature`: the number of node features.
  - `edge_feature`: the number of edge features.
  - `node_count`: the number of nodes.
  - `edge_count`: the number of edges.
  - `num_classes`: the number of node classes.
  - `density`: the density of the graph, calculated by the NetworkX library.
  - `connected_components`: the set of vertices in a graph that are linked to each other by paths.
  - `average_degree`: the average degree of all the nodes in the graph.
  - `average_clustering_coefficient`: the degree to which nodes in a graph tend to cluster together.
  - `average_degree_centrality`: the average fraction of connected nodes across the whole graph.
  - `average_eigenvector_centrality`: the average influence of all nodes in the graph.
  - `local_average_clustering_coefficient`: the average clustering coefficient of the subgraphs.

```json
{
    "name": "cora",
    "type": "dataset",
    "property": {
        "node_feature": 1433,
        "local_average_clustering_coefficient": nan,
        "average_clustering_coefficient": 0.24067329850193728,
        "average_degree_centrality": 0.0014399999126942077,
        "local_average_betweenness_centrality": 0.0,
        "edge_count": 5278,
        "num_classes": 7,
        "density": 0.0014399999126942077,
        "average_eigenvector_centrality": 0.0047865456098027765,
        "local_average_closeness_centrality": 1.0,
        "node_count": 2708,
        "average_degree": 3.8980797636632203,
        "connected_components": 78,
        "local_graph_diameter": 1.0,
        "edge_feature": 0,
        "local_average_shortest_path_length": 1.0,
        "description": 'The Cora dataset is a citation network of 2,708 machine-learning papers, organized into seven distinct
classes. These papers are interlinked by 5,429 citations, forming a directed graph that maps out how papers cite each other.'
    },
    "id": 0
}
```

Figure 7: The structure of the `Data` entity in the KBG.

- – `local_average_betweenness_centrality`: the average sum of the fraction of all-pairs shortest paths that pass through a node of the subgraphs.
  - – `local_graph_diameter`: the length of the shortest path between the most distanced nodes of the subgraphs.
  - – `local_average_closeness_centrality`: the average reciprocal of the sum of the shortest path distances from a node to all other nodes of the subgraphs.
  - – `description`: the semantic description of the dataset.
  - –

  The example is shown in Fig. 7.
- • `Model`: the model contains the following attributes.:
  - – `num_post_layers`: the number of embedding layers before passing the raw node features into the GNN.
  - – `num_prev_layers`: the number of embedding layers after receving the hidden embeddings from the GNN.
  - – `dimension`: the dimension of the hidden embeddings.
  - – `dropout`: the drop out rate of the model.
  - – `learning_rate`: the learning rate of the model.
  - – `weight_decay`: the weight decay during optimization.
  - – `num_epoch`: the number of epochs of training the model.

  And the example is shown in Fig. 8a.

As for the relations, the most important relation is `hasPerformance` that records the performance information of the KBG. An example is shown in Fig. 8b and its attributes include:

- • `perf(acc_test: ..., acc_valid: ..., loss: ...)`: the performance information of the testing and validation accuracies and the loss on the test set is recorded in the form of a dictionary.
- • `latency`: the model training time.
- • `para`: the number of parameters of the optimized model.
- • `task`: the task for the corresponding performance record.

The entity and relation statistics of the extracted KBG is shown in Tab. 6 and Tab. 7. Please note that the search space $\mathbb{F}^c$ contains 26,206 variants of model architectural designs, but considering the fine-grained hyper-parameters of different models, we finally have 211,669 model instances in the

```
{
    "name": "00",
    "type": "model",
    "property": {
        "num_prev_layers": 0,
        "num_post_layers": 1,
        "dimension": 256,
        "dropout": 0.7,
        "learning_rate": 0.1,
        "weight_decay": 0.0005,
        "num_epoch": 400
    },
    "id": 17
}
```

```
{
    "source_entity": 0,
    "target_entity": 17,
    "relation": "has_performance",
    "property": {
        "perf": {
            "acc_test": 0.797,
            "acc_valid": 0.7733333333333334,
            "loss": 1.234
        },
        "latency": 0.005922238032023112,
        "para": 1.736711,
        "task": "Node Classification"
    },
    "id": 2328395
}
```

(a) The example of `model` entity.  (b) The example of `hasPerformance` relation.

Figure 8: Effect of performance threshold $\tau$ on $\epsilon$, $\Delta$ and performance.

Table 6: The entity statistics of KBG.

| Entity Category | Entity Type | Num Instances |
|---|---|---|
| Dataset | Dataset | 9 |
| | Domain | 4 |
| | Heterogeneity | 2 |
| Model | Model | 211,669 |
| | Optimizer | 2 |
| | GPU | 2 |
| | CPU | 2 |
| | OS | 2 |
| | Structure Topology | 9 |
| | Layer Structure | 9 |
| Common | Task | 1 |
| | Modality | 1 |
| Total Number | 12 | 211,712 |

KBG, which leads to the increased number facts. However, in the actual model design retrieval, only the architectural design is considered, thus the valid number of models is still 26,206.

Subsequent to Nas-Bench-Graph, many methods are developed to improve the GNN design effectiveness via proposing a more effective search space. Early efforts explicitly incorporate link information into designing (Di et al., 2021; WANG et al., 2021), enhancing node classification and link prediction. Subsequent studies introduced fine-tuning search spaces, improving pre-trained GNN adaptation (WANG et al., 2024), while others developed data-adaptive GNNs that dynamically adjust receptive fields based on graph properties (Wang et al., 2023b). There are also AutoGNN methods explored message-passing function search on knowledge graphs, enhancing model expressiveness but remaining constrained to specific KG structures (Di & Chen, 2023). As the search space design is an orthogonal task to our work, we do not compare our method with these methods in the main text and stick to Nas-Bench-Graph.

## A.4 BENCHMARK DATA INFORMATION

The datasets used in the benchmark data are: Cora (Sen et al., 2008), Citeseer (Sen et al., 2008), Pubmed (Sen et al., 2008), CS (Shchur et al., 2018), Physics (Shchur et al., 2018), Photo (Shchur et al., 2018), Computers (Shchur et al., 2018), Arxiv (Hu et al., 2020), and Proteins (Hu et al., 2020). The statistics of the datasets are shown in Tab. 8. On the other hand, the models used in

Table 7: The relation statistics of KBG.

| Relation Category | Relation Type | Num Instances |
|---|---|---|
| Dataset | hasDomain | 9 |
| | hasHeterogeneity | 9 |
| | hasTask | 9 |
| | hasModality | 9 |
| Model | hasOptimizer | 211,669 |
| | hasGPU | 211,669 |
| | hasCPU | 211,669 |
| | hasOS | 211,669 |
| | hasArchGraph | 211,669 |
| | hasLayerOperator_1 | 211,669 |
| | hasLayerOperator_2 | 211,669 |
| | hasLayerOperator_3 | 211,669 |
| | hasLayerOperator_4 | 211,669 |
| | hasTask | 211,669 |
| | hasModality | 211,669 |
| Dataset-Model | hasPerformance | 211,669 |
| Total Number | 16 | 2,540,064 |

the benchmark data have 9 unique topologies with 4 nodes in the directed acyclic graph (DAG) and 9 different layer operations, ['GCN', 'GAT', 'SAGE', 'Skip', 'GIN', 'Cheb', 'FC', 'ARMA', 'Graph'], in each node.

Table 8: List of datasets used in the benchmark data.

| Dataset | Nodes | Edges | Classes | Description |
|---|---|---|---|---|
| Cora | 2,708 | 5,429 | 7 | Citation |
| Citeseer | 3,327 | 4,732 | 6 | Citation |
| Pubmed | 19,717 | 44,338 | 3 | Citation |
| CS | 18,333 | 81,894 | 15 | Coauthor |
| Physics | 34,493 | 247,962 | 5 | Coauthor |
| Photo | 7,487 | 119,043 | 8 | Shopping |
| Computers | 13,381 | 245,778 | 10 | Social |
| Arxiv | 169,343 | 1,166,243 | 40 | Citation |
| Proteins | 132,534 | 39,561,252 | 112 | Protein |

## A.5 FULL EXPERIMENTS ON THE MODEL RETRIEVAL SCORE

The full results of the Tab. 2 with standard deviation is provided in Tab. 9. Because the similarity based retrieval methods always find the same model across different runs, the standard deviation is not reported.

## A.6 ADDITIONAL EXPERIMENTS ON THE IMPACT OF $\tau$, $\epsilon$, $\Delta$

In addition to the results in Sec. 7.3, we provide the case studies on the other two datasets Cora and Physics in Fig. 9 and Fig. 10. Across the three datasets, we can observe a increasing trend of $\Delta$ with $\epsilon$, which further validates the observation in Sec. 7.3. A higher $\tau$ will generally result in a higher transferred performance on all three datasets, which is also supportive to our observation.

Table 9: The performance of different baseline methods on the datasets used with standard deviation.

| Model | Cora | Citeseer | PubMed | CS | Physics | Photo | Computer | arXiv |
|---|---|---|---|---|---|---|---|---|
| GCN | 80.97±0.39 | 69.90±1.26 | 77.46±0.61 | 88.65±0.57 | 90.85±1.20 | 89.44±0.48 | 83.16±0.55 | 71.08±0.16 |
| GAT | 80.83±0.47 | 70.70±0.71 | 75.93±0.26 | 88.72±0.73 | 89.47±1.14 | 89.93±1.75 | 81.35±1.26 | 71.24±0.10 |
| SAGE | 79.47±0.31 | 66.13±0.90 | 75.50±1.14 | 87.81±0.18 | 91.43±0.29 | 88.29±1.03 | 81.46±0.73 | 70.78±0.17 |
| GIN | 79.77±0.38 | 63.30±1.26 | 76.74±0.86 | 81.08±3.09 | 86.67±0.86 | 87.37±1.01 | 73.95±0.16 | 61.33±0.70 |
| ChebNet | 79.40±0.57 | 67.03±1.02 | 75.13±0.49 | 89.50±0.36 | 89.75±0.87 | 86.65±0.77 | 79.10±2.26 | 70.87±0.10 |
| ARMA | 78.33±0.69 | 66.20±0.75 | 75.00±0.51 | 89.87±0.35 | 88.88±1.09 | 86.55±3.35 | 78.47±0.57 | 70.87±0.17 |
| k-GNN | 78.06±0.47 | 30.97±3.56 | 75.38±0.97 | 83.81±0.58 | 88.98±0.54 | 86.45±0.21 | 76.31±1.34 | 63.18±0.38 |
| Random | 77.87±2.41 | 66.64±1.32 | 74.16±1.68 | 81.78±9.41 | 90.59±0.94 | 89.04±2.55 | 76.61±3.56 | 68.93±1.82 |
| EA | 78.23±1.04 | 66.40±2.63 | 72.88±2.11 | 87.03±2.64 | 88.07±2.41 | 87.30±1.38 | 77.56±6.42 | 68.28±2.95 |
| RL | 73.44±8.11 | 65.35±2.40 | 75.44±1.24 | 86.17±5.09 | 88.15±4.24 | 89.48±1.35 | 77.70±3.07 | 68.00±4.71 |
| GNAS | 78.55±1.20 | 63.25±5.87 | 73.04±1.64 | 86.04±7.88 | 89.54±1.52 | 87.27±2.96 | 70.96±9.66 | 69.94±1.71 |
| Auto-GNN | 78.58±2.18 | 65.60±2.69 | 76.07±0.77 | 89.06±0.42 | 89.26±1.51 | 89.34±1.75 | 77.49±3.41 | 67.62±1.72 |
| GPT4GNAS | 78.50±0.37 | 67.46±0.76 | 73.89±0.86 | 89.26±0.38 | 89.44±1.94 | 89.12±2.26 | 77.21±5.26 | 68.98±1.22 |
| GHGNAS | 79.13±0.45 | 67.35±0.44 | 74.90±0.57 | 89.15±0.81 | 88.94±2.57 | 89.42±1.99 | 77.04±3.96 | 69.66±1.28 |
| DesiGNN | 80.31±0.00 | 69.20±0.16 | 76.60±0.00 | 89.64±0.08 | 92.10±0.00 | 91.19±0.00 | 82.20±0.00 | 71.50±0.00 |
| Kendall | 67.73 | 69.20 | 71.80 | 88.56 | 91.56 | 88.90 | 76.85 | 71.49 |
| Overlap | 79.36 | 67.30 | 71.80 | 88.56 | 89.95 | 90.37 | 76.85 | 71.68 |
| Ours | 82.53 | 69.20 | 76.53 | 89.32 | 90.34 | 90.37 | 76.61 | 71.68 |

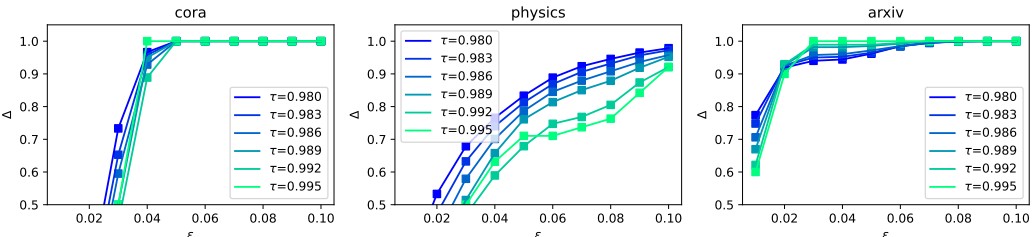

Figure 9: Effect of $\tau$ on $\epsilon$ and $\Delta$.

## A.7 COMPLEX DATASET SIMILARITY COMPARISON

In this section, we extend our experiment on different dataset similarity metrics. In the main context, we compared several simple simialrities, and here we propose include a more complex similarity metric, the Personalized PageRank (PPR) algorithm, to further evaluate the dataset similarity metrics.

When implementing PPR, the original similarities $\mathcal{S}(D^b, D^u)$ calculated by the dataset features serve as the initial weights of edges. Considering that the unseen dataset $D^u$ do not have model performance information, we neglect the performance relations between $\mathbb{D}^b$ and benchmark models $\mathbb{M}^b$ as they are less helpful for dataset similarity.

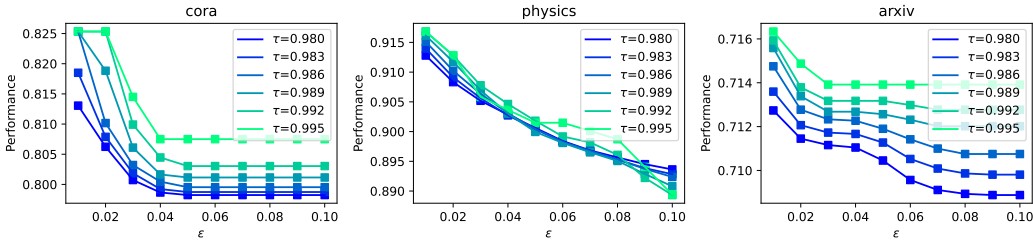

Figure 10: Effect of $\tau$ on $\epsilon$ and performance.

Table 10: The RLS of different similarity metrics in initial model suggestion.

| Metric | Cora | Citeseer | PubMed | CS | Phys. | Photo | Comp. | Arxiv |
|---|---|---|---|---|---|---|---|---|
| Simple-L1 | 0.452 | 0.126 | 0.003 | 0.020 | 0.013 | 0.310 | 0.025 | 0.014 |
| Simple-L2 | 0.498 | 0.064 | 0.012 | 0.007 | 0.040 | **0.328** | 0.036 | 0.012 |
| Simple-Ours | 0.435 | 0.340 | 0.004 | 0.025 | 0.002 | 0.252 | 0.043 | 0.027 |
| Complex-PPR | **0.853** | **0.768** | **0.278** | **0.340** | **0.344** | 0.316 | **0.485** | **0.493** |

The experiment results are shown in Tab.10. A higher RLS indicates that the dataset similarity metric can better find a similar dataset. The best values are shown in **bold** and second best in underline.

Results show that, our metric obtains a better performance than other naive metrics for its surpression on the overly large feature differences. When further considering the edge information among datasets, the PPR algorithm significantly improves the dataset similarity quality over the naive methods. This further validates the essence of constructing the knowledge benchmark graph. However, the linearity is still not high enough (maximally 1), indicating that there is still room for improvement in the dataset similarity metric design.

## A.8 DETAILS IN COMBINING KBG WITH LLM

When combining KBG and LLM to suggest model designs, we position LLM as a model design optimization tool that could understand and leverage the information retrieved from KBG. As we are agnoistic to the LLM designs, we use the GPT-4 model as the representative LLM in our experiments. The detailed procedure is as follows:

1. Given a KBG and unseen dataset $D^u$, find similar datasets $\mathbb{D}^c$ to $D^u$ with dataset similarity score $\mathcal{S}(\cdot, \cdot)$.

2. Retrieve promising candidate models $\mathbb{F}^c$ with the model relevance score $\mathcal{R}(\cdot)$ based on $\mathbb{D}^c$.

3. Provide information of $D^u$, $\mathbb{D}^c$, $\mathbb{F}^c$ and historical results and instruct the LLM to infer (design a new model based on $\mathbb{F}^c$) or select (pick the most suitable model from $\mathbb{F}^c$) a model $F'$ for $D^u$.

4. Evaluate and record the performance of $F'$ on $D^u$.

5. Retrieve another set of $\mathbb{F}^c$ from KBG based on: 1) relevance $\mathcal{R}(\cdot)$ to $D^u$ and 2) similar model architectures to $F'$.

6. Repeat steps 3-5 until the computation budget is exhausted or the performance of $F'$ is satisfactory.

When retrieving the candidate models $\mathbb{F}^c$ at the step 5, we combine the model relevance score $\mathcal{R}(\cdot)$ with the model archietectural similarity (noted as $\mathcal{W}(\cdot)$) to find more suitable models. To calculate $\mathcal{W}(\cdot)$, we first transform the topology and layer operations of the benchmark models into binary vectors $\mathbf{w}$ of 0s and 1s. Each element of $\mathbf{w}$ stands for a part of arthitecture design, and 1 means that benchmark models have the same architecture as the current model design $F'$. Then, we average the elements in $\mathbf{w}$ as the model architecture similarity $\mathcal{W}(F, F')$ for $F \in \mathbb{F}^b, F \neq F'$. Finally, we use $\beta \mathcal{R}(F; D^u) + (1 - \beta)\mathcal{W}(F, F')$ as the final model relevance for selecting next set of candidates, where $\beta$ is the weight hyper-parameter and set to 0.8 in our experiments.

When designing the prompts for LLMs, we adopt an intuitive and effective prompting practice without excessive tuning because we aim to validate the effectiveness of our KBG as extra knowledge in assisting LLMs design models. Following the prior examples in Wang et al. (2023a); Dong et al. (2023); Wang et al. (2024), the prompts are organized into five textual components:

1. Task description

2. Model space description

3. Optimization trajectory

4. Candidate models from KBG

5. Role-play instruction for Infer/Select strategies

These textual segments are updated (specifically, components (3) and (4)) and combined into a single prompt template during each iteration before sending to GPT-4 API for the response. Especially, when receiving the response from LLM, we require the LLM to provide the structure topology and layer operations of the recommended model, along with the recommendation reason.

Afterwards, we extract the structure topology and layer operations to build the model automatically. Then, the recommended model is tested on the unseen dataset to obtain the performance feedback, which will be further combined with the model details and appended to the optimization history in the prompt for the next iteration. More importantly, the recommended model will serve as the anchor for retrieving the next round of similar models from KBG, thereby improving the model performance over time.

Throughout the process, the only constraint applied to LLMs is that they cannot recommend a model already tested in the optimization history, which can be enforced by proper instruction. Please note that our KBG can capture the similarity between datasets and models, which is fundamentally a filtering mechanism that only retrieves the most relevant and effective knowledge before sending them to LLMs for reference. Therefore, we did not apply further filtering to LLM's suggestions so that the effectiveness of our KBG in this study could be directly observed.

## A.9    CASE STUDY OF MODEL RETRIEVAL ON THE KBG

To help understand our process, we first breifly introduce the pipeline of retrieving initial model designs from KBG for unseen dataset below.

1. Given a KBG with becnhmark datasets $\mathbb{D}^b$ and unseen dataset $D^u$, we calculate the dataset similarity score $\mathcal{S}(D^b, D^u)$ between $D^u$ and each $D^b \in \mathbb{D}^b$ based on their statistical features.

2. Select the similar datasets $\mathbb{D}^c = \{\mathcal{S}(D^b, D^u) \geq \delta | D^b \in \mathbb{D}^b\}$ to $D^u$ up to a threshold $\delta$.

3. Calculate the relevance score $\mathcal{R}(\cdot)$ of the benchmark models $\mathbb{F}^b$ towards $D^u$ based on their performances on $\mathbb{D}^c$.

4. Select the candidate models $\mathbb{F}^c = \{\mathcal{R}(F^b) \geq \tau | F^b \in \mathbb{F}^b\}$ for $D^u$ up to a threshold $\tau$.

5. Recommend $\mathbb{F}^c$ to $D^u$.

For a case study, suppose cora dataset is the unseen dataset $D^u$, the more concrete steps are:

1. Calculate the dataset similarity between cora and other benchmark datasets, whose similarities are listed in Tab. 11.

2. Then set the threshold $\delta$ to 0.7 and obtain the candidate datasets $\mathbb{D}^c = \{\text{Comp.}, \text{Photo}\}$.

3. Calculate the model relevance $\mathcal{R}(F; D^u)$ of the benchmark models towards cora based on their performances on $\mathbb{D}^c = \{\text{computers}, \text{photo}\}$. Suppose a model $F$ has the reletiave performance of 0.8 on computers and 0.7 on photo, then

$$\mathcal{R}(F; D^u) = 0.8 \times 0.735902 + 0.7 \times 0.724709 \approx 1.095.$$

4. Select models with relevance score higher than $\tau$.

5. Recommend the selected models to cora.

Table 11: The similarity between cora and other benchmark datasets.

| Dataset | Comp. | Photo | Citeseer | Pubmed | CS | Phys. | arxiv |
|---|---|---|---|---|---|---|---|
| Similarity | 0.735902 | 0.724709 | 0.69855 | 0.698388 | 0.688489 | 0.665778 | 0.647834 |

## A.10    EXPERIMENTS ON APPLYING KBG TO IMAGE CLASSIFICATION BENCHMARK

In the main experiments, we demonstrated the effectiveness of our framework in the graph domain. In this section, we further validate its generalization ability by applying it to the computer vision

domain. This demonstrates that our framework is extensible across different data modalities and model designs.

This generalizability is achieved through our comprehensive ontology, which specifies how to organize dataset features, model architectures, and performance information into the Knowledge Benchmark Graph (KBG) without relying on domain-specific knowledge. Additionally, both our dataset similarity metric and model relevance metric depend solely on the constructed KBG, ensuring the framework's adaptability across domains.

We conduct experiments on the NATS-Bench dataset (Dong et al., 2021), a benchmark in the computer vision domain for image classification tasks. NATS-Bench contains performance records for three image datasets and 15,645 unique topological architectures.

**Dataset Features and KBG Construction**   For this experiment, we extract features of the image datasets by transforming the statistical information of raw images into vectors and summarizing each statistical feature across all samples. The remaining construction procedures follow our standard framework.

Using the NATS-Bench data and our KBG ontology, we constructed a CV-KBG (Knowledge Benchmark Graph for Computer Vision). The CV-KBG was then used to calculate dataset similarity scores as defined in Sec. 5. The results are shown in Tab. 12.

Table 12: Dataset similarity scores for the datasets ImageNet16-120, Cifar10 and Cifar100 that are used in NATS-Bench.

|  | ImageNet16-120 | Cifar10 | Cifar100 |
|---|---|---|---|
| **ImageNet16-120** | 1.000000 | 0.177472 | 0.311604 |
| **Cifar10** | 0.177472 | 1.000000 | 0.510925 |
| **Cifar100** | 0.311604 | 0.510925 | 1.000000 |

The results align with intuitive expectations: CIFAR-10 and CIFAR-100 are closely related, while ImageNet16-120 is more similar to CIFAR-100 than to CIFAR-10, reflecting the greater diversity of image classes in CIFAR-100.

**Model Retrieval for Unseen Datasets**   Next, we simulated unseen datasets by excluding one dataset and retrieving the best models from the remaining similar datasets for the unseen dataset, whose results are shown in Tab. 13.

Table 13: Comparison between optimal and retrieved models for unseen datasets. The test accuracy of the retrieved models was compared with the accuracy of the optimal models, with rankings of the retrieved models shown in parentheses.

|  | ImageNet16-120 | Cifar10 | Cifar100 |
|---|---|---|---|
| Optimal | 38.27 | 89.16 | 61.18 |
| Retrieved | 38.27 (1/15625) | 88.96 (6/15625) | 60.44 (14/15625) |

These results validate that our retrieval method is effective in identifying high-performing models for unseen datasets in the computer visions domain. This demonstrates the generalization ability and practical utility of our framework.

