# OpenReview forum: "Structuring Benchmark into Knowledge Graphs to Assist Large Language Models in Retrieving and Designing Models"
_ICLR.cc/2025/Conference — ICLR 2025 Poster_

### Official Review · Reviewer_JG3d · 2024-10-25

**Soundness:** 3
**Presentation:** 3
**Contribution:** 3
**Rating:** 8
**Confidence:** 3

**Summary:**

- This paper proposes a novel AutoML framework with LLM for GNN design. In this paper, the authors construct a knowledge benchmark graph to inform the LLM with more domain knowledge about GNN architecture and design new metrics to guide the knowledge selection.

**Strengths:**

- The idea of this paper of paper is more innovative, combining Knowledge Grap, LLM and AutoML.
- The authors have done sufficient data preparation, method design and experiments around this idea.

**Weaknesses:**

- There are some labeling errors in Table2, e.g., the optimal result on the Citeseer dataset appears on the GAT but is not bold.
- The experiments in the current article stop at the GNN domain and are only oriented to the node classification task, which has some limitations in the scope of application. And the title gives the impression that the authors' approach is oriented to a variety of tasks in generalized scenarios. I think KBG can be considered to be applied on top of more heterogeneous tasks, such as other tasks in the field of graph learning, or even out-of-domain experiments such as CV/NLP that require the GNN method to verify the effectiveness of the method. Further, the design of the model can also be not limited to the GNN model.

**Questions:**

- Which LLM do you employ in the experiments? I can not get the corresponding information after reading Section 5.1
- The naming of knowledge benchmark graph might be modified. I think the current name misleads people into thinking this is a new KG benchmark.
- See Weaknesses

---

> ### Author Response · Authors · 2024-11-21
> **Reply to Reviewer JG3d**
>
> We sincerely thank the reviewer for the insightful comments and constructive feedback.
> We appreciate the reviewer for recognizing our **novel idea and sufficient method design**.
>
> We have carefully considered the comments and revised the paper accordingly, revisions are in blue color. In the following reply, indecies of definitions, sections and equations refer to the revised manuscript.
>
> ***Progress towards finishing the rebuttal:***
>
> [⬛⬛⬛⬛⬛⬛⬛⬛⬛⬛] JG3d-W1: Finished.
>
> [⬛⬛⬛⬛⬛⬛⬛⬛⬛⬛] JG3d-W2: Finished.
>
> [⬛⬛⬛⬛⬛⬛⬛⬛⬛⬛] JG3d-Q1: Finished.
>
> [⬛⬛⬛⬛⬛⬛⬛⬛⬛⬛] JG3d-Q2: Finished
>
> ## **JG3d-W1**: Labeling errors in Table 2
>
> ## **Reply to JG3d-W1**:
>
> We appreciate the reviewer for pointing out the labeling errors, we have **corrected the notations in Tab. 2**.
>
> ---
>
> ## **JG3d-W2**: Limited experiments in the GNN domain
>
> ## **Reply to JG3d-W2**:
>
> Our framework is extensible to different data modalities and different application tasks.
> Because we propose a comprehensive ontology that clearly specifies how to organize the dataset features, model architectures, and the performance information into a knowledge benchmark graph (KBG) without requiring domain-specific knowledge.
> And our dataset similarity metric and model relevance metric are only dependent on the constructed KBG.
>
> To better illustrate that our framework is extensible to various scenarios, we conduct an experiment on the NATS-Bench dataset [1].
> NATS-Bench is a benchmark dataset on computer vision domain, containing performance records on 3 image datasets and 15,625 CNN topological architectures.
> Especially, we extract the features of image datasets by transforming the statistical information of raw images into a set of vectors and summarizing each statistical feature across all samples.
> The other construction procedure remains unchanged.
>
> Currently, the experiment is ongoing, and we will provide preliminary results within the rebuttal period.
>
> [1] 2021 TPAMI - NATS-Bench: Benchmarking NAS Algorithms for Architecture Topology and Size
>
> ---
>
> ## **JG3d-Q1**: Lack of detailed description of large models
>
> ## **Reply to JG3d-Q1**:
>
> **We have polished Sec. 7.6 and added Appendix A.6 for better clarification.**
>
> We utilized GPT-4 in our experiments, in line with other LLM-based AutoML baselines in our experiment.
>
> When combining KBG and LLM to suggest model designs, we position LLM as a model design optimization tool that could understand and leverage the information retrieved from KBG.
> The detailed procedure is as follows:
>
> 1. Given a KBG and unseen dataset $D^u$, find similar datasets $\mathbb{D}^c$ to $D^u$ with dataset similarity score $\mathcal{S}(\cdot, \cdot)$.
> 2. Retrieve promising candidate models $\mathbb{F}^c$ with the model relevance score $\mathcal{R}(\cdot)$ based on $\mathbb{D}^c$.
> 3. Provide information of $D^u$, $\mathbb{D}^c$, $\mathbb{F}^c$ and historical results and instruct the LLM to infer (design a new model based on $\mathbb{F}^c$) or select (pick the most suitable model from $\mathbb{F}^c$) a model $F'$ for $D^u$.
> 4. Evaluate and record the performance of $F'$ on $D^u$.
> 5. Retrieve another set of $\mathbb{F}^c$ from KBG based on: 1) relevance $\mathcal{R}(\cdot)$ to $D^u$ and 2) similar model architectures to $F'$.
> 6. Repeat steps 3-5 until the computation budget is exhausted or the performance of $F'$ is satisfactory.
>
> ---
>
> ## **JG3d-Q2**: Misleading name of Knowledge Benchmark Graph
>
> ## **Reply to JG3d-Q2**:
>
> We greatly thank the reviewer for pointing out the potential confusion caused by the name "Knowledge Benchmark Graph".
> We will polish our name in the final version.

---

> > ### Comment · Reviewer_JG3d · 2024-11-22
> > **Thank you for your rebuttal**
> >
> > Dear Author:
> >
> > Thank you for your rebuttal. It address my concerns. Thus, I will keep my score.

---

> > > ### Author Response · Authors · 2024-11-27
> > >
> > > We are glad to know that your concerns are addressed. We greatly thank you for your efforts in reviewing our paper and providing instructive feedback.
> > >
> > > Continuing our **Reply to JG3d-W2**, we have now obtained the primary experimental results on extending our method to computer vision domain, which will be described below. The related results are also included in **Appendix A.8** of the revised manuscript.
> > >
> > > ---
> > >
> > > **KBG Construction and Dataset Similarity Calculation**
> > >
> > > Using the NATS-Bench dataset and our KBG ontology, we constructed a **CV-KBG (Knowledge Benchmark Graph for Computer Vision)**.
> > > The CV-KBG is then used to calculate dataset similarity scores as defined in **Sec. 5**. The results are shown below:
> > >
> > > |  | ImageNet16-120 | CIFAR-10 | CIFAR-100 |
> > > |---:|---:|---:|---:|
> > > | **ImageNet16-120** | 1.000000 | 0.177472 | 0.311604 |
> > > | **CIFAR-10** | 0.177472 | 1.000000 | 0.510925 |
> > > | **CIFAR-100** | 0.311604 | 0.510925 | 1.000000 |
> > >
> > > The results align with intuitive expectations: CIFAR-10 and CIFAR-100 are closely related, while ImageNet16-120 is more similar to CIFAR-100 than to CIFAR-10 for having more diverse classes of images.
> > >
> > > **Model Retrieval for Unseen Datasets**
> > >
> > > Next, we simulate unseen datasets by excluding one dataset and retrieving the best models from the remaining similar datasets for the unseen dataset. The testing classification accuracy of the retrieved models was compared with the accuracy of the optimal models, with rankings of the retrieved models shown in parentheses:
> > >
> > > |  | ImageNet16-120 | CIFAR-10 | CIFAR-100 |
> > > |---:|---:|---:|---:|
> > > | **Optimal** | 38.27 | 89.16 | 61.18 |
> > > | **Retrieved** | 38.27 (1/15625) | 88.96 (6/15625) | 60.44 (14/15625) |
> > >
> > > These results validate that our retrieval method is effective in identifying high-performing models for unseen datasets for image classification tasks under the computer vision domain. This experiment further demonstrates the generalization ability of our framework to various data modalities.

---

### Official Review · Reviewer_eVNQ · 2024-11-02

**Soundness:** 3
**Presentation:** 3
**Contribution:** 3
**Rating:** 6
**Confidence:** 4

**Summary:**

This paper attempts to construct a graph that stores all the existing datasets, models, and model performance on datasets for future research and development endeavors. Based on the constructed datasets, the authors try to propose some metrics to evaluate the similarity between datasets and the effectiveness of the models retrieved on an unseen dataset. The experimental results show that such a graph is benefical for the development of AutoML.

**Strengths:**

The idea is interesting and the motivation is convincing.
The implementation of this idea is relatively complete, including the graph construction process, the design of enhancing generalization ability on incorporating unseen datasets, and the retrieval mechanism of existing model candidates.

**Weaknesses:**

The work is full of engingeering skills while lack some academic insights. For example, controling and varying the hyperparameters (e.g., delta, the size of datasets, epsilon, etc.) bring limited insight. Perhaps a case study is required to illustrate how the algorithm succeeds to retrieve a good model according to a given unseen-yet-similar dataset. There should be more deeper insights and factors beyond the similarity of datasets, such as the underlying common research issues. What features should the algorithm capture and consider?
The scenario is relatively limited. Authors conduct experiments merely in GNN domain. It’s unclear whether such an effort could generlize to other ML methods, which makes the contribution of this paper vague. I suggest conducting a small amount of experimental evidence to demonstrate the generalization ability of this work, which could make it more promising and convincing.

**Questions:**

Please refer to the weakness.

---

> ### Author Response · Authors · 2024-11-21
> **Reply to Weakness 1 of Reviewer eVNQ (1 / 2)**
>
> We sincerely thank the reviewer for the insightful comments and constructive feedback.
> We appreciate the reviewer for recognizing **our novel idea and the completeness of our method**.
>
> We have carefully considered the comments and revised the paper accordingly, revisions are in blue color. In the following reply, indices of definitions, sections and equations refer to the revised manuscript.
>
> ***Progress towards finishing the rebuttal:***
>
> [⬛⬛⬛⬛⬛⬛⬛⬛⬛⬛] eVNQ-W1: Finished.
>
> [⬛⬛⬛⬛⬛⬛⬛⬛⬛⬛] eVNQ-W2: FInished.
>
> ## **eVNQ-W1**: Lack some academic insights
>
> ## **Reply to eVNQ-W1**:
>
> **We have revised our problem definition in Sec. 3 and evaluation on the dataset similarity in Sec. 6 to better present the insights of our work**
>
> We appreciate the reviewer’s feedback and would like to emphasize the unique contributions of this work, particularly in addressing the open challenge of leveraging benchmark knowledge for automated model retrieval and design.
>
> 1. **Structured Knowledge Representation with the Knowledge Benchmark Graph (KBG):**
>     The Knowledge Benchmark Graph (KBG) offers a novel, structured approach to organizing benchmark data, encoding relationships between datasets, models, and their performance. This enables efficient retrieval and reasoning, addressing the limitations of brute-force model search in existing AutoML methods.
> 2. **Dataset Similarity as a Core Insight:**
>     By introducing and formalizing dataset similarity  $\mathcal{S}(\cdot,\cdot)$ , we provide a principled way to guide model selection based on performance relationships. This metric aligns with the intuition that similar datasets prefer similar models and offers a practical tool for improving model transferability.
> 3. **Comprehensive Validation:**
>     The framework is thoroughly validated through experiments across various datasets and conditions, demonstrating its robustness and practical utility. This includes evaluating the effects of dataset similarity, hyperparameters, and KBG structure, providing empirical support for the proposed methods.
>
> These contributions establish a strong foundation for further exploration.
>
> ---
>
> ## **eVNQ-W1.1**: Controling and varying the hyperparameters bring limited insight
>
> ## **Reply to eVNQ-W1.1**:
>
> We acknowledge that hyperparameter variation, such as  $\Delta$, $\epsilon$ and $\mathbb{D}^c$, primarily serves to validate the robustness of our framework and help understand how the dataset similarity can be evaluated.
>
> While the current analysis focuses on the empirical behavior of the framework, we recognize the importance of linking these results to deeper insights. For example, although the model architectures do not have obvious relationships with their performance on different datasets, our experiments demonstrate that such relationships exist (Like in Figure 4 (a), better performing models tend to maintain its advantage across datasets). This observation motivates further research into the underlying factors that influence model performance across datasets.
>
> For another example, simply aggregating information from KBG with our simple dataset similarity and model relevance metrics (results in Tab. 2 and Tab. 4) already yields promising results. This intrigues future work to further explore more effective and efficient retrieval methods for KBG to enhance the recommendation process.

---

> ### Author Response · Authors · 2024-11-21
> **Reply to Weakness 1 of Reviewer eVNQ (2 / 2)**
>
> ## **eVNQ-W1.2**: A case study can better illustrate how the algorithm succeeds
>
> ## **Reply to eVNQ-W1.2**:
>
> **We have added the case study in Appendix A.7**
>
> We appreciate the suggestion that a case study can better demonstrate our method.
> To help understand our process, we first briefly introduce the pipeline of retrieving initial model designs from KBG for the unseen dataset below.
>
> 1. Given a KBG with becnhmark datasets $\mathbb{D}^b$ and unseen dataset $D^u$, we calculate the dataset similarity score $\mathcal{S}(D^b, D^u)$ between $D^u$ and each $D^b \in \mathbb{D}^b$ based on their statistical features.
> 2. Select the similar datasets $\mathbb{D}^c = \{\mathcal{S}(D^b, D^u) \geq \delta | D^b \in \mathbb{D}^b\}$ to $D^u$ up to a threshold $\delta$.
> 3. Calculate the relevance score $\mathcal{R}(\cdot)$ of the benchmark models $\mathbb{F}^b$ towards $D^u$ based on their performances on $\mathbb{D}^c$.
> 4. Select the candidate models $\mathbb{F}^c = \{\mathcal{R}(F^b) \geq \tau | F^b \in \mathbb{F}^b\}$ for $D^u$ up to a threshold $\tau$.
> 5. Recommend $\mathbb{F}^c$ to $D^u$.
>
> For a case study, suppose cora dataset is the unseen dataset $D^u$, the more concrete steps are:
>
> 1. Calculate the dataset similarity between cora and other benchmark datasets, whose similarities are listed below.
>
> |   Dataset  | computers |   photo  | citeseer |  pubmed  |    cs    |  physics |   arxiv  |
> |:----------:|:---------:|:--------:|:--------:|:--------:|:--------:|:--------:|:--------:|
> | Similarity |  0.735902 | 0.724709 |  0.69855 | 0.698388 | 0.688489 | 0.665778 | 0.647834 |
>
> 1. Then set the threshold $\delta$ to 0.7 and obtain the candidate datasets $\mathbb{D}^c = \{\text{computers}, \text{photo}\}$.
> 2. Calculate the model relevance $\mathcal{R}(F;D^u)$ of the benchmark models towards cora based on their performances on $\mathbb{D}^c = \{\text{computers}, \text{photo}\}$. Suppose a model $F$ has the relative performance of 0.8 on computers and 0.7 on photo, then $\mathcal{R}(F;D^u) = 0.8 \times 0.735902 + 0.7 \times 0.724709 \approx 1.095$.
> 3. Select models with relevance scores higher than $\tau$.
> 4. Recommend the selected models to cora.
>
> ---
>
> ## **eVNQ-W1.3**: There should be more insights and research issues beyond the similarity of datasets, like what features the algorithm should capture and consider.
>
> ## **Reply to eVNQ-W1.3**:
>
> Our work focuses more on dataset similarity because our method holds the assumption that similar datasets prefer similar models. This assumption sets the effectiveness of dataset similarity at the core of our framework. However, we agree that there are more insights or research issues that could be considered in the recommendation process, which we have presented in **Reply to eVNQ-W1.1**.
>
> Currently, our method does not assign weights in utilizing the raw features and the features are directly extracted using graph statistical measurements.
> But we do agree that how to weight the dataset and model features in calculating the dataset similarity and model relevance is an interesting research issue. Because the importance of different features may vary under different application tasks. Regarding the wide possibility of unseen datasets, building an automatic feature selection mechanism is essential to improve the recommendation performance.

---

> ### Author Response · Authors · 2024-11-21
> **Reply to Weakness 2 of Reviewer eVNQ**
>
> ## **eVNQ-W2**: Application scenario is relatively limited
>
> ## **Reply to eVNQ-W2**:
>
> Our framework is extensible to different data modalities and different designs of models.
> Because we propose a comprehensive ontology that clearly specifies how to organize the dataset features, model architectures, and performance information into a knowledge benchmark graph (KBG) without requiring domain-specific knowledge.
> And our dataset similarity metric and model relevance metric are only dependent on the constructed KBG.
>
> To better illustrate that our framework is extensible to various scenarios, we are conducting an extensive experiment on the NATS-Bench dataset [1].
> NATS-Bench is a benchmark dataset on the computer vision domain, containing performance records on 3 image datasets and 15,625 unique topological architectures.
> Especially, we extract the features of image datasets by transforming the statistical information of raw images into a set of vectors and summarizing each statistical feature across all samples.
> The other construction procedure remains unchanged.
>
> Currently, the experiment is ongoing, and we will provide preliminary results within the rebuttal period.
>
> [1] 2021 TPAMI - NATS-Bench: Benchmarking NAS Algorithms for Architecture Topology and Size

---

> > ### Author Response · Authors · 2024-11-27
> >
> > Continuing our **Reply to eVNQ-W2**, we have now obtained the primary experimental results on extending our method to computer vision domain, which will be described below. The related results are also included in **Appendix A.8** of the revised manuscript.
> >
> > **We look forward to having discussions with you.**
> >
> > ---
> >
> > **KBG Construction and Dataset Similarity Calculation**
> >
> > Using the NATS-Bench dataset and our KBG ontology, we constructed a **CV-KBG (Knowledge Benchmark Graph for Computer Vision)**.
> > The CV-KBG is then used to calculate dataset similarity scores as defined in **Sec. 5**. The results are shown below:
> >
> > |  | ImageNet16-120 | CIFAR-10 | CIFAR-100 |
> > |---:|---:|---:|---:|
> > | **ImageNet16-120** | 1.000000 | 0.177472 | 0.311604 |
> > | **CIFAR-10** | 0.177472 | 1.000000 | 0.510925 |
> > | **CIFAR-100** | 0.311604 | 0.510925 | 1.000000 |
> >
> > The results align with intuitive expectations: CIFAR-10 and CIFAR-100 are closely related, while ImageNet16-120 is more similar to CIFAR-100 than to CIFAR-10 for having more diverse classes of images.
> >
> > **Model Retrieval for Unseen Datasets**
> >
> > Next, we simulate unseen datasets by excluding one dataset and retrieving the best models from the remaining similar datasets for the unseen dataset. The testing classification accuracy of the retrieved models was compared with the accuracy of the optimal models, with rankings of the retrieved models shown in parentheses:
> >
> > |  | ImageNet16-120 | CIFAR-10 | CIFAR-100 |
> > |---:|---:|---:|---:|
> > | **Optimal** | 38.27 | 89.16 | 61.18 |
> > | **Retrieved** | 38.27 (1/15625) | 88.96 (6/15625) | 60.44 (14/15625) |
> >
> > These results validate that our retrieval method is effective in identifying high-performing models for unseen datasets for image classification tasks under the computer vision domain. This experiment further demonstrates the generalization ability of our framework to various data modalities.

---

> ### Comment · Reviewer_eVNQ · 2024-11-27
>
> The article is indeed very inspiring, but it still has some shortcomings at present. Even after the rebuttal, the issues were not completely resolved, so I will maintain my current rating.

---

> > ### Author Response · Authors · 2024-11-30
> >
> > We sincerely appreciate your recognition of our work’s inspiration and your thoughtful feedback.
> >
> > Regarding your remaining concern about our academic insights, we would like to further highlight our contribution to AutoML. When designing models for unseen datasets, training cost is a major bottleneck in improving design efficiency. Orthogonal to current AutoML approaches that focus on enhancing model-design optimizers, our work introduces a new direction to reduce training costs by leveraging benchmark knowledge to approximate model performance on unseen datasets (more detailed discussion is presented in **Reply to NmMt-W1.1**). As we continue our study, many research issues are worth investigating, such as how to design effective dataset similarity metrics and how to extend our work to measure similarity between different tasks.
> >
> > Once again, we highly value your suggestions and will certainly incorporate them into our future research.

---

### Official Review · Reviewer_zioe · 2024-11-03

**Soundness:** 3
**Presentation:** 3
**Contribution:** 2
**Rating:** 5
**Confidence:** 3

**Summary:**

The authors proposes a new solution for AutoML. The authors design a knowledge graph that contains information on data, model and performance. With the knowledge graph, the method uses similarity score of data and relevance score of model to suggest best model on unseen data.

In experiments, the authors construct the knowledge graph with graph datasets and GNN architectures. The result show that the method author proposes achieves the best result on 3/8 tasks. However, with the assistance of LLM, the model is able to achieve best result on 5/8 tasks.

**Strengths:**

The paper writing is clear. The formulation is very straight-forward. The authors use data similarity score and model relevance score to infer the best potential model on unseen data.
The authors provide extensive experiments comparing to SOTA methods.

**Weaknesses:**

1. The data similarity score the authors propose is too simple. The authors focus their study on GNN, but the similarity score does not involve any relational information on the edges.
2. The model relevance score is confusing. The name sounds like it look for architectural similarity between models. But in reality it's the model's historical performance.
2. The experiment section that involves LLM is very vague to me. There's is not explanation on exactly how LLM infer or select the models.

In summary, the method basically proposes models based on the following two ideas: a) if the node features are similar 2) if the model has historical performance.

**Questions:**

1. LLM assisted method achieves best performance but there is no detailed explanation or design. For example, you can answer the following questions:
    (a)What prompts or instructions were given to the LLM?
    (b)How were the LLM's outputs processed or integrated into the model selection process?
    (c)Were there any constraints or filtering applied to the LLM's suggestions?
2. It's good to include more complicated design of the two scores. For example, incorporate edge-level information in data similarity score calculation and model architectural information, like the number of convolutional layers, in model relevance calculation.

---

> ### Author Response · Authors · 2024-11-20
> **Overall reply and progress summary in finishing the rebuttal.**
>
> We sincerely thank the reviewer for the insightful comments and constructive feedback.
> We appreciate the reviewer for recognizing our **clear presentation and thorough experimental evaluations**.
>
> We have carefully considered the comments and revised the paper accordingly, revisions are in blue. In the following reply, indices of definitions, sections, and equations refer to the revised manuscript.
>
> ***Progress towards finishing the rebuttal:***
>
> [⬛⬛⬛⬛⬛⬛⬛⬛⬛⬛] zioe-W1: Finished.
>
> [⬛⬛⬛⬛⬛⬛⬛⬛⬛⬛] zioe-W2: Finished.
>
> [⬛⬛⬛⬛⬛⬛⬛⬛⬛⬛] zioe-W3: Finished.
>
> [⬛⬛⬛⬛⬛⬛⬛⬛⬛⬛] zioe-Q1: Finished.
>
> [⬛⬛⬛⬛⬛⬛⬛⬛⬛⬛] zioe-Q2: Finished.

---

> ### Author Response · Authors · 2024-11-20
> **Reply to Weakness 1, 2 & 3 of Reivewer zioe**
>
> ## **zioe-W1**: Data similarity score is too simple
>
> ## **Reply to zioe-W1**:
>
> We want to emphasize that how to design the dataset similarity score is **not the focus of our work**, as noted in **Sec. 5**.
> Our contributions are to identify the novel problem in AutoML of efficient model selection for unseen datasets using existing benchmark knowledge (**clarified in Sec. 3**), construct a novel knowledge graph ontology for organizing benchmark data (**described in Sec. 4**), and provide clear problem definition and comprehensive evaluation metric for designing future dataset similarity scores (**clarified in Sec. 6**).
> We have **revised Sec. 3 to Sec. 6** to better clarify our contribution and the necessity of our design.
>
> Given our method framework, we intentionally designed a simple dataset similarity score to validate our novel problem while aligning with our objective of efficient model retrieval. Designing an effective and efficient dataset similarity score is indeed an open challenge, and our work offers a practical starting point for future research. We acknowledge that leveraging the relational information in the KBG is a plausible way to enhance the quality of dataset similarity score, and have provided a related experiment in **Reply to zioe-Q2** to support subsequent studies.
>
> ---
>
> ## **zioe-W2**: The name model relevance score is confusing
>
> ## **Reply to zioe-W2**:
>
> **We have revised Sec. 5 for a clearer introduction to the model relevance score.**
>
> After identifying benchmark datasets similar to the unseen dataset  $D^u$ , our goal is to retrieve models from the Knowledge Benchmark Graph (KBG) that are likely to perform well on  $D^u$ . This retrieval process can be intuitively understood as evaluating how relevant the models are to  $D^u$ . Accordingly, we name this retrieval score $\mathcal{R}(\cdot)$ “model relevance” to properly reflect its objective.
>
> When calculating  $\mathcal{R}(\cdot)$, we exclude architectural information because there are no existing performance relationships between benchmark models and  $D^u$. As a result, model architectures provide limited insight into determining relevance to  $D^u$. Instead, based on the assumption that “similar datasets prefer similar models,” the historical performance of models on benchmark datasets serves as a proxy for their effectiveness on  $D^u$, provided the datasets are sufficiently similar. Therefore, we define  $\mathcal{R}(\cdot)$  using the historical performance of a model on similar benchmark datasets to  $D^u$, weighted by the dataset similarities.
>
> ---
>
> ## **zioe-W3**: Vague description of LLM+KBG method
>
> ## **Reply to zioe-W3**:
>
> **We have polished the description of the LLM+KBG method in Sec. 7.6 and provided more details in Appendix A.6.**
>
> When combining KBG and LLM to suggest model designs, we position LLM as a model design optimization tool that could understand and leverage the information retrieved from KBG.
> The detailed procedure is as follows:
>
> 1. Given a KBG and unseen dataset $D^u$, find similar datasets $\mathbb{D}^c$ to $D^u$ with dataset similarity score $\mathcal{S}(\cdot, \cdot)$.
> 2. Retrieve promising candidate models $\mathbb{F}^c$ with the model relevance score $\mathcal{R}(\cdot)$ based on $\mathbb{D}^c$.
> 3. Provide information of $D^u$, $\mathbb{D}^c$, $\mathbb{F}^c$ and historical results and instruct the LLM to infer (design a new model based on $\mathbb{F}^c$) or select (pick the most suitable model from $\mathbb{F}^c$) a model $F'$ for $D^u$.
> 4. Evaluate and record the performance of $F'$ on $D^u$.
> 5. Retrieve another set of $\mathbb{F}^c$ from KBG based on: 1) relevance $\mathcal{R}(\cdot)$ to $D^u$ and 2) similar model architectures to $F'$.
> 6. Repeat steps 3-5 until the computation budget is exhausted or the performance of $F'$ is satisfactory.
>
> Specifically, **Infer** refers to the prompt instruction for LLM to design a new model based on the observed patterns among $\mathbb{F}^c$, while in the **Select** process, LLM is asked to select the most suitable model from $\mathbb{F}^c$ based on the context.

---

> ### Author Response · Authors · 2024-11-20
> **Reply to Question 1 of Reviewer zioe**
>
> ## **zioe-Q1**: Detailed explanation of LLM-assisted method
>
> ## **Reply to zioe-Q1**:
>
> **We provided more method details in Appendix A.6.**
>
> We adopt an intuitive and effective prompting practice without excessive tuning because we aim to validate the effectiveness of our KBG as extra knowledge in assisting LLMs design models.
> Continuing from the example process in **Reply to zioe-W3**, our answers to the questions are listed as follows:
>
> (a) Following [1], [2], [3], our prompt design is organized into five textual components:
>
> 1. Task description
> 2. Model space description
> 3. Optimization trajectory
> 4. Candidate models from KBG
> 5. Role-play instruction for Infer/Select strategies
>
> These textual segments are updated (specifically, components (3) and (4)) and combined into a single prompt template during each iteration before sending to GPT-4 API for the response.
>
> (b) The output of LLM, phrased using OpenAI’s structured output, contains three fields (using NAS-Bench-Graph as an example):
>
> 1. Structure topology of the recommended model
> 2. Layer operations of the recommended model
> 3. Reason for the recommendation.
>
> After receiving the response, we extract the structure topology and layer operations to build the model automatically. Then, the recommended model is tested on the unseen dataset to obtain the performance feedback, which will be further combined with the model details and appended to the optimization history in the prompt for the next iteration. More importantly, the recommended model will serve as the anchor for retrieving the next round of similar models from KBG, thereby improving the model performance over time.
>
> (c) The only constraint applied to LLMs is that they cannot recommend a model already tested in the optimization history, which can be enforced by proper instruction. Please note that our KBG can capture the similarity between datasets and models, which is fundamentally a filtering mechanism that only retrieves the most relevant and effective knowledge before sending them to LLMs for reference. Therefore, we did not apply further filtering to LLM’s suggestions so that the effectiveness of our KBG in this study could be directly observed.
>
> The corresponding details are added to the revised manuscript in Appendix A.6.
>
> [1] 2023 - Graph Neural Architecture Search with GPT-4
>
> [2] 2023 - Heterogeneous Graph Neural Architecture Search with GPT-4
>
> [3] 2024 - Computation-friendly Graph Neural Network Design by Accumulating Knowledge on Large Language Models

---

> ### Author Response · Authors · 2024-11-25
> **Reply to Question 2 of Reviewer zioe**
>
> ## **zioe-Q2**: It's good to include more complicated design of the two scores.
>
> ## **Reply to zioe-Q2**:
>
> We appreciate the suggestions for incorporating more complex information in the data similarity score and model relevance score.
>
> ### **More complex dataset similarity**
>
> We realize more dataset similarity metrics should be included for a more comprehensive ablation study.
>
> Thus, we propose to apply Personalized PageRank (PPR) algorithm to the relations among unseen dataset $D^u$ and benchmark datasets $\mathbb{D}^b$ to better identify the similar benchmark datasets to $D^u$.
> The original similarities $\mathcal{S}(D^b, D^u)$ calculated by the dataset features serve as the initial weights of edges.
> Considering that the unseen dataset $D^u$ does not have model performance information, we neglect the performance relations between $\mathbb{D}^b$ and benchmark models $\mathbb{M}^b$ as they are less helpful for dataset similarity.
>
> The experiment results are shown below, where the reported values are the relevance linearity score (RLS) proposed in Sec. 6.2 for evaluating the dataset similarity metrics. A higher RLS indicates that the dataset similarity metric can better find a similar dataset.
> The best values are shown in **bold** and second best in *italic*.
>
> Results show that, when further considering the edge information, the PPR algorithm significantly improves the dataset similarity quality over the naive methods.
> This further validates the essence of constructing the knowledge benchmark graph.
> However, the linearity is still not high enough (maximally 1), indicating that there is still room for improvement in the dataset similarity metric design.
>
> |Metric|Cora|Citeseer|PubMed|CS|Physics|Photo|Computers|Arxiv|
> |---|---|---|---|---|---|---|---|---|
> |Simple-L1|0.452|0.126|0.003|0.020|0.013|0.310|0.025|0.014|
> |Simple-L2|*0.498*|0.064|*0.012*|0.007|*0.040*|**0.328**|0.036|0.012|
> |Simple-Ours|0.435|*0.340*|0.004|*0.025*|0.002|0.252|*0.043*|*0.027*|
> |Complex-PPR|**0.853**|**0.768**|**0.278**|**0.340**|**0.344**|*0.316*|**0.485**|**0.493**|
>
> ### **More complex model relevance score**
>
> **We included the utilization of architectural information during model refinement in Appendix A.6.**
>
> Following the response to **Reply to zioe-W2**, architectural information is unsuitable for the initial stage of model retrieval.
> However, when KBG is combined with LLM to iteratively refine models, the suggested models offer a chance to include architectural information.
> And we have already combined the model-architecture similarity with historical-performance similarity to give a more comprehensive recommendation, which is described in step 5 of the example in **Reply to zioe-W3**.
>
> The architectural similarity $\mathcal{W}(\cdot)$ is calculated by the average number of similar architecture design elements between benchmark models and the currently suggested model $F'$.
> And we use $\beta \mathcal{R}(F; D^u) + (1-\beta) \mathcal{W}(F, F')$ as the score to rank benchmark models and pick the candidate models $\mathbb{F}^b$ for further recommendation, where $\beta$ is the weight hyperparameter and set to 0.8 in practice.

---

> > ### Author Response · Authors · 2024-11-27
> >
> > Dear reviewer zioe, we appreciate your insightful comments and valuable suggestions for improving our work. In response to your concerns, we have updated the experiments to incorporate a more complex dataset similarity metric, as detailed in **Reply to zioe-Q2**.
> >
> > Furthermore, we have conducted an additional experiment to validate the extensibility of our approach to other data modalities than graphs. The results of this experiment are provided below and included in **Appendix A.8** of the revised manuscript.
> >
> > **We look forward to having discussions with you.**
> >
> > ---
> >
> > To demonstrate the extensibility of our framework to various scenarios, we conducted experiments on the **NATS-Bench dataset** [1], a benchmark in the computer vision domain for image classification tasks. NATS-Bench contains performance records for three image datasets and 15,645 unique topological architectures.
> > The primary experimental results are described below.
> >
> > **KBG Construction and Dataset Similarity Calculation**
> >
> > Using the NATS-Bench dataset and our KBG ontology, we constructed a **CV-KBG (Knowledge Benchmark Graph for Computer Vision)**. Dataset features were extracted by converting the statistical information of raw images into feature vectors, with each statistical property summarized across all samples. The rest of the KBG construction process adhered to our standard framework.
> >
> > The CV-KBG is then used to calculate dataset similarity scores as defined in **Sec. 5**. The results are shown below:
> >
> > |  | ImageNet16-120 | CIFAR-10 | CIFAR-100 |
> > |---:|---:|---:|---:|
> > | **ImageNet16-120** | 1.000000 | 0.177472 | 0.311604 |
> > | **CIFAR-10** | 0.177472 | 1.000000 | 0.510925 |
> > | **CIFAR-100** | 0.311604 | 0.510925 | 1.000000 |
> >
> > The results align with intuitive expectations: CIFAR-10 and CIFAR-100 are closely related, while ImageNet16-120 is more similar to CIFAR-100 than to CIFAR-10 for having more diverse classes of images.
> >
> > **Model Retrieval for Unseen Datasets**
> >
> > Next, we simulated unseen datasets by excluding one dataset and retrieving the best models from the remaining similar datasets for the unseen dataset. The testing classification accuracy of the retrieved models was compared with the accuracy of the optimal models, with rankings of the retrieved models shown in parentheses:
> >
> > |  | ImageNet16-120 | CIFAR-10 | CIFAR-100 |
> > |---:|---:|---:|---:|
> > | **Optimal** | 38.27 | 89.16 | 61.18 |
> > | **Retrieved** | 38.27 (1/15625) | 88.96 (6/15625) | 60.44 (14/15625) |
> >
> > These results validate that our retrieval method is effective in identifying high-performing models for unseen datasets for image classification tasks under the computer vision domain. This experiment further demonstrates the generalization ability of our framework to various data modalities.
> >
> > [1] 2021 TPAMI - NATS-Bench: Benchmarking NAS Algorithms for Architecture Topology and Size

---

> ### Author Response · Authors · 2024-11-30
>
> Dear Reviewer,
>
> Thank you for your efforts in reviewing our paper and providing thoughtful feedback.
> In case you missed our previous responses, we briefly summarize our responses to your comments to help you catch up quickly.
>
> First, we want to clarify a misunderstanding about our primary contribution.
> Our work aims to **identify and validate a novel problem in AutoML**, specifically the challenge of efficient model selection for unseen datasets using existing knowledge. While we propose initial methods to address this problem, our main contribution lies in bringing attention to the problem itself, formating a clear definition and demonstrating its importance through empirical validation.
>
> Then, regarding to your two major concerns, our responses are summarized below:
>
> ---
>
> ### **1. Dataset Similarity and Model Relevance Scores**
>
> We appreciate your suggestions on enhancing the dataset similarity and model relevance scores. Our current simple designs for $\mathcal{S}(\cdot)$ and $\mathcal{R}(\cdot)$ are intentional, serving as baseline methods to initiate discussion on this new problem. We aimed for efficiency and practicality, enabling quick model recommendations without extensive computations. Incorporating complex metrics, such as edge-level information or architectural details, could indeed improve accuracy but may not align with our goal of proposing an accessible starting point for future research.
>
> To address your concerns and facilitate further studies, we have:
>
> - **Clarified the purpose of our simple scoring methods** in **Reply to zioe-W1** and **Sec. 5**, emphasizing their role as foundational tools rather than final solutions.
> - **Added a more complex dataset similarity metric** using the Personalized PageRank algorithm in **Reply to zioe-Q2**, demonstrating potential improvements and encouraging exploration of more sophisticated methods.
> - **Clarified the difficulty in using architectural information** in the initial model retrieval due to the lack of performance relationships, as explained in **Reply to zioe-W2**. And described how architectural information is incorporated during model refinement in **Reply to zioe-Q2** and **Appendix A.6**.
>
> ---
>
> ### **2. Details on LLM-Assisted Model Selection**
>
> Regarding the LLM-assisted method, we acknowledge that the initial description lacked detail. Our intent was to illustrate how Large Language Models (LLMs) can be integrated into the model selection process as a complementary tool rather than the primary focus.
>
> In response, we have **provided a detailed description** on the LLM integration in **Reply to zioe-W3**, **Reply to zioe-Q1**, and **Appendix A.6**, outlining the prompts used, how outputs are processed, and any constraints applied.
>
> ---
>
> We are grateful for your insights, which have helped us improve the clarity and depth of our manuscript.
> Please feel free to reach out with any additional questions or suggestions.

---

### Official Review · Reviewer_NmMt · 2024-11-04

**Soundness:** 3
**Presentation:** 3
**Contribution:** 3
**Rating:** 6
**Confidence:** 2

**Summary:**

This paper introduces a graph dataset that helps connect datasets, models, and model performance, making it easier for machine learning systems to automatically find the best model architecture for a specific dataset. Since real-world datasets are often new and unseen, the authors create a method to measure how relevant different datasets are to each other, which helps in sharing knowledge between them. This method allows the system to use information from existing benchmark data, ensuring that high-performing models can still be applied to new datasets. Additionally, the authors present a new metric that focuses on the most useful insights, which makes the model selection process even better. In their experiments, they test this approach on various datasets to show how effective and efficient it is, highlighting its potential to improve model design and performance in real-world situations.

**Strengths:**

This paper has several notable strengths that enhance its contribution to the field of automated machine learning.

1. The introduction of a comprehensive graph dataset models the relationships between datasets, models, and performance. This structured resource simplifies the model selection process for researchers and practitioners.
2. The theoretical framework is well-articulated and provides a solid basis for the proposed methods. This enhances the credibility of the approach and demonstrates a deep understanding of the principles involved.
3. The experiments conducted are thorough and well-executed, testing the methods across various datasets. These results provide strong empirical support for the authors’ theoretical claims.
4. The research has significant implications for automated machine learning (AutoML), allowing for the automatic identification of optimal model architectures. This capability can reduce the time and expertise required for model design, making machine learning more accessible.

Overall, the paper effectively combines a valuable dataset, strong theoretical foundations, and solid experimental validation, positioning it as a promising contribution to AutoML. Its findings could lead to further advancements in automated processes for model development.

**Weaknesses:**

The theoretical explanations in the paper could be improved with additional background to aid reader comprehension.
1. For example, in Definition 1, it would be useful for the authors to provide an overview of existing problem formulations in model transfer or AutoML to better contextualize their approach.
2. Explaining the motivation for using a probability lower bound in Definition 1 and its relevance to practical model transfer would clarify this choice.
3. It would also be helpful to indicate whether this problem formulation is novel or based on existing methods, and if it is novel, to discuss the advantages it brings over previous formulations.

In Section 4.3, the intuition behind the transferability score could be further clarified.
1. A conceptual explanation of what the transferability score represents in practical terms would be beneficial, along with a small example or illustration to demonstrate how it is calculated and interpreted.
2. Additionally, comparing this score with existing metrics for evaluating model transfer effectiveness could further clarify its utility.

Furthermore, the paper’s discussion on integrating Large Language Models (LLMs) into the proposed framework could be more comprehensive, as it is currently quite brief.
1. The authors might expand on the specific role of LLMs in their approach, detailing how they interact with the Knowledge Benchmark Graph and contribute to model selection or adaptation.
2. Examples illustrating the LLMs' role in the process would be helpful, as well as a discussion of any potential challenges in integrating LLMs and how these are addressed.
3. Lastly, comparing this approach to other recent methods that incorporate LLMs for AutoML or model selection would provide a useful context for the reader.

**Questions:**

please refer to weaknesses

---

> ### Author Response · Authors · 2024-11-20
> **Overall reply**
>
> We sincerely thank the reviewer for the insightful comments and constructive feedback.
> We appreciate the reviewer for recognizing our contribution to **reducing time and expertise required for designing models** by proposing a **comprehensive graph dataset modeling data-model-performance relationships, well-articulated theoretical framework, and thorough experimental evaluations**.
>
> We have carefully considered the comments and revised the paper accordingly, where revisions are in blue. In the following reply, indices of definitions, sections, and equations refer to the revised manuscript.

---

> ### Author Response · Authors · 2024-11-20
> **Reply to Weakness 1 of Reviewer NmMt (1 / 2)**
>
> ##  **NmMt-W1**: The theoretical explanations could be improved with additional background to aid reader comprehension.
>
> ## **Reply to NmMt-W1**:
>
>
> We have **reorganized the manuscript** to present our theory and methods more clearly and intuitively in Sec. 3 to Sec. 6.
> - Sec. 3: Summarizes the overall problem definition, providing a foundation for the subsequent sections.
> - Sec. 4: Introduces the construction of the Knowledge Benchmark Graph (KBG) and highlights its advantages in organizing benchmark data.
> - Sec. 5: Details our proposed model retrieval method based on the KBG.
> - Sec. 6: Focuses on the core of our framework, dataset similarity. We formally define the problem of designing a dataset similarity score (formally Def. 1) and propose a novel similarity evaluation metric.
>
> ---
>
> ## **NmMt-W1.1**: An overview of existing problem formulations in model transfer or AutoML is helpful
>
> ## **Reply to NmMt-W1.1**:
>
> We have included the **problem definition of AutoML in Sec. 2** and provided a comparison between our problem formulation and existing AutoML and model transfer methods in **Sec. 3**.
>
> Briefly, our method focuses on retrieving models for unseen datasets using benchmark data, avoiding the computational overhead of model training. **In contrast**:
>
> - Existing **AutoML** methods require training models as part of the search process.
> - Existing **model transfer** methods address data scarcity by reusing a specific model trained on a source domain for a target domain.
>
> To compare our problem with existing AutoML and transfer learning methods with greater detail and clarity, we first briefly present our definition.
> Given an unseen dataset $D^u$, searching for an optimal model $F^*$ from possible models $\mathbb{F}^b$ that has the best performance $\mathcal{M}(F;D^u)$ can be formulated as:
>
> $$
> F^* = \arg\max_{F \in \mathbb{F}^b} \mathcal{M}(F;D^u).
> $$
>
> To address the efficiency issue in searching models, we adopt the intuition that *similar datasets prefer similar models* and leverage information from benchmark data to simplify the problem, which contains a set of models $\mathbb{F}^b$ with known performances on datasets  $\mathbb{D}^b$.
> Based on the benchmark data, we transform the problem into evaluating the similarity between $D^u$ and $\mathbb{D}^b$ and retrieving the most relevant model from $\mathbb{F}^b$ for $D^u$. Our problem is defined as:
>
> $$
> \max_{F \in \mathbb{F}^b} \mathcal{M}(F;D^u) \propto \max_{F \in \mathbb{F}^b} \mathcal{M}(F;D^b)\cdot\mathcal{S}(D^b;D^u), \ \ \ \ \text{Reply-Eq}.(1)
> $$
>
> where $\mathcal{S}(\cdot, \cdot)$ is the similarity score between datasets.
>
> ### **Comparison with AutoML**
>
> The AutoML methods focus on designing good model searching algorithms $\pi (\cdot)$ that samples models from the model space $\mathbb{F}^b$ to find $F^*$, which is formulated as:
>
> $$
> F^* = \arg\max_{F \sim \pi(F)} \text{E}[\mathcal{M}(F;D^u)],
> $$
>
> where $\text{E}[\cdot]$ is the expectation over the model performances.
> The optimization process of the problem is formulated as:
>
> $$
> \nabla_{\pi} \text{E}[\mathcal{M}(F;D^u)] \approx \frac{1}{|\mathbb{F}^c|} \sum_{F^i \in \mathbb{F}^c} \mathcal{M}(F^i;D^u) \ \nabla_{\pi} \log \pi (F^i),
> $$
>
> where $\mathbb{F}^c$ is the candidate model set for improving $\pi(F)$.
>
> During optimization, a larger $|\mathbb{F}^c|$ increases the model recommendation quality from $\pi(F)$.
> However, enlarging $\mathbb{F}^c$ also **increases the computation overhead** of training models to obtain $\mathcal{M}(F,D^u)$.
> Additionally, $\pi(F)$ faces the **cold start issue** to search models from scratch for unseen datasets.
>
> **Our problem differs from AutoML methods** in reducing the cost of obtaining $\mathcal{M}(F, D^u)$ by approximating it with retrieved information from benchmark data, leading to a zero training cost for model recommendation.
> Besides, the output of our problem provides a good initial model for AutoML methods to ease the cold start issue.
>
> ### **Comparison with Model transfer**
>
> The existing model transfer methods focus on transferring the model parameters $\omega$ from the source domain $D^s$ to the previously unseen domain $D^u$ to reduce the data sparsity issue. The model transfer problem can be formulated as:
>
> $$
> F^*_{\omega} = \arg\min_{\omega} \mathcal{L}(F^s_{\omega};D^u).
> $$
>
> where $F^s_{\omega}$ is the model trained on $D^s$ and $\mathcal{L}(\cdot)$ is the transfer learning loss function.
> Designing an effective $\mathcal{L}(\cdot)$ to better adapt $F^s_{\omega}$ to $D^u$ is the major focus of model transfer methods.
> **Different from them**, while our method focuses on efficiently recommending model designs to $D^u$ based on the benchmark model performances.

---

> ### Author Response · Authors · 2024-11-20
> **Reply to Weakness 1 of Reviewer NmMt (2 / 2)**
>
> ## **NmMt-W1.2**: Explain the motivation of defining the problem with a probability lower bound and its relation to practical model transfer
>
> ## **Reply to NmMt-W1.2**:
>
> We have revised the presentation of dataset similarity theory definition in **Sec. 6** to enhance clarity and motivation.
>
> Designing an effective dataset similarity metric is an open challenge with various potential approaches.
> Relying solely on the transfer performance of **a single model** can lead to significant bias and fail to capture the broader relationship between datasets.
> To address this, we consider the transfer performances of **a set of candidate models** $\mathbb{F}^c$ to provide a more comprehensive evaluation of dataset similarity.
>
> Intuitively, similar datasets should support the successful transfer of a large number of models with high performances.
> This relationship can be quantified by a higher probability  $\Delta$  of successful transfers within a small performance drop tolerance  $\epsilon$.
> To reflect this, our definition of dataset similarity incorporates a probability lower bound, which provides a minimum level of confidence for successful model transfer within the specified transfer loss tolerance.
>
> In practice, the model retrieval performance heavily depends on whether the models of benchmark datasets perform well on unseen datasets.
> Our probability lower bound definition can robustly reflect such characteristic and provide guidance for future similarity metric designs.
>
> ---
>
> ## **NmMt-W1.3**: Indicate whether this problem formulation is novel or based on existing methods, and specify its advantages
>
> ## **Reply to NmMt-W1.3**:
>
> Our problem of suggesting promising models for unseen datasets by **retrieving from benchmark data without training** is novel problem and distinct from existing AutoML and model transfer methods.
> In particular, our definition and evaluation of the dataset similarity metric in Section 6 are key contributions to the field.
>
> There are prior AutoML methods [1] [2] that utilize the benchmark data to train a surrogate model for predicting the performance of candidate models on unseen datasets.
> These surrogate models are trained by using the dataset and model features as the input and the model performances as the ground truth.
> However, as discussed in Sec. 2 of the paper, the **computation cost of building surrogate models is still unavoidable**.
>
> Moreover, these methods also operate under the assumption that “similar datasets prefer similar models,” which makes their effectiveness heavily dependent on a robust dataset similarity measurement. Unfortunately, they **lack a formal definition of dataset similarity and overlook the exploration of how dataset similarity impacts the performance of recommended models**. This gap limits their comprehensiveness and highlights the novelty of our approach.
>
> In contrast, our method bypasses the need for training by directly retrieving relevant models from benchmark data. This design makes our approach significantly more efficient and scalable to much larger benchmark datasets, setting it apart from surrogate model-based methods.
>
>
> [1] 2013 ICML - Collaborative hyperparameter tuning
>
> [2] 2021 NeurIPS - Automatic Unsupervised Outlier Model Selection

---

> ### Author Response · Authors · 2024-11-20
> **Reply to Weakness 2 of Reviewer NmMt**
>
> ## **NmMt-W2**: In Section 4.3, the intuition behind the transferability score could be further clarified.
>
> ## **Reply to NmMt-W2**:
>
> We have **polished Sec. 6.1** to offer a clearer background of the transferability score and added a conceptual explanation in **Sec. 6.2** to address the whole Weakness 2.
>
> In our paper, we define two datasets as similar if many models perform well on both. To evaluate different dataset similarity metrics, it is crucial to obtain a ground truth for how models transfer between datasets. To this end, based on our definition of dataset similarity in Sec. 6.1,  we introduce the transferability score  $\mathcal{T}(\cdot)$ to measure the success rate of high-performing transfers between benchmark and unseen datasets.
>
> ---
>
> ## **NmMt-W2.1**: A conceptual explanation of the transferability score with an example would be beneficial
>
> ## **Reply to NmMt-W2.1**:
>
> Simply saying, given a set of candidate models $\mathbb{F}^c$ selected from benchmark dataset $D^b$ and transferred to unseen dataset $D^u$, a higher transferability score $\mathcal{T}(\mathbb{F}^c)$ reflects a higher success rate $\Delta$ of transferring $\mathbb{F}^c$ to $D^u$ within a smaller performance drop tolerance $\epsilon$.
> (Please refer to **Reply to NmMt-W1.2** for a detailed introduction of $\Delta$ and $\epsilon$.)
>
> An example of estimating $\hat{\mathcal{T}}(\mathbb{F}^c)$ is presented as follows.
>
> Suppose when transferring $\mathbb{F}^c$ from $D^b$ to $D^u$, the curve between the success transfer rate $\Delta$ and the performance drop tolerance $\epsilon$ is empirically evaluated on $\epsilon = \{0.01, 0.02, 0.03, 0.04, 0.05\}$ as follows:
>
> | $\epsilon$ | 0.01 | 0.02 | 0.03 | 0.04 | 0.05 |
> |------------|------|------|------|------|------|
> | $\Delta$   | 15%  | 40%  | 70%  | 80%  | 90%  |
>
> Then the estimated transferability score $\hat{\mathcal{T}}(\mathbb{F}^c)$ is calculated as:
>
> $\hat{\mathcal{T}}(\mathbb{F}^c) = \frac{1}{5} \sum^{0.05}_{\epsilon=0} \Delta(\epsilon) = \frac{1}{5} $ (15\% + 40\%+ 70\% + 80\% + 90\%) = 0.59.
>
> If $\mathbb{F}^c$ has a larger transferrability, it means that $\Delta$ gains a higher value at each $\epsilon$, e.g. $\Delta$ increases from 15% to 20% at $\epsilon =0.01$.
> Ideally, when $\mathbb{F}^c$ is perfectly transferrable, $\Delta$ should be 100% at all $\epsilon$, meaning that $\mathbb{F}^c$ perform equally well both $D^b$ and $D^u$.
>
> ---
>
> ## **NmMt-W2.2**: Comparing this score with existing metrics for evaluating model transfer effectiveness could further clarify its utility
>
> ## **Reply to NmMt-W2.2**:
>
> Given the novelty of our problem, there lack of related metric designs for evaluating model transfer effectiveness.
> So we compare our metric with two intuitive methods:
>
> **Performance-difference based metric $\mathcal{T}_{diff}(\cdot)$** measures how performances of $\mathbb{F}^b$ on $D^u$ differ from those on $D^b$:
>
>  $\mathcal{T}_{diff}(\mathbb{F}^c) = \frac{1}{| \mathbb{F}^c |} \sum ( \mathcal{M} (F;D^u) - \mathcal{M} (F;D^b) )$,
>
> where the summation is for $F \in \mathbb{F}^c$.
> $\mathcal{T}_{diff}(\mathbb{F}^c)$ becomes larger when $\mathbb{F}^c$ performs better on $D^u$.
> The average performance difference is **vulnerable to outliers** when a few models perform significantly better or worse than the others.
> Thus fails to reflect the overall landscape of model transferability.
>
> **Rank-correlation based metric $\mathcal{T}_{rank}(\cdot)$** measures how the performance rankings of $\mathbb{F}^c$ on $D^u$ and $D^b$ are correlated, which may take the form of Spearman's rank correlation coefficient.
> $\mathcal{T}_{rank}(\mathbb{F}^c)$ becomes larger when $\mathbb{F}^c$ orders more similarly in performance on $D^u$ and $D^b$.
> Unfortunately, $\mathcal{T}\_{rank}(\cdot)$ **overlooks the scale of performance differences**.
> When two models order differently on $D^u$ and $D^b$ but have a neglectable performance difference, they should still be consistent, yet $\mathcal{T}\_{rank}(\cdot)$ will give a low score.
>
> **Compared with the two metrics**, our metric reduces the effect from outliers by focusing on the successful transfer rate and handles performance-difference scales by considering multiple performance drop thresholds.
> Thus, our metric is more robust to different situations and can more honestly reflect the model transferability.

---

> ### Author Response · Authors · 2024-11-20
> **Reply to Weakness 3 of Reviewer NmMt (1 / 2)**
>
> ## **NmMt-W3**: The paper’s discussion on integrating LLMs into the proposed framework could be more comprehensive
>
> ## **Reply to NmMt-W3**:
>
> **We have polished the corresponding experiment description in Sec. 7.6 and provided more experiment details in Appendix A.6.**
>
> ---
>
> ## **NmMt-W3.1**: The authors might expand on the specific role of LLMs in their approach
>
> ## **Reply to NmMt-W3.1**:
>
> When combining KBG and LLM to suggest model designs, we position LLM as a model design optimization tool that could understand and leverage the information retrieved from KBG.
> Following our discussion about the AutoML in **Reply to NmMt-W1.1**, LLM is an alternative of $\nabla_{\pi} \log \pi (F^i)$ that refine the model designs.
> The detailed procedure is as follows:
>
> 1. Given a KBG and unseen dataset $D^u$, find similar datasets $\mathbb{D}^c$ to $D^u$ with dataset similarity score $\mathcal{S}(\cdot, \cdot)$.
> 2. Retrieve promising candidate models $\mathbb{F}^c$ with the model relevance score $\mathcal{R}(\cdot)$ based on $\mathbb{D}^c$.
> 3. Provide information of $D^u$, $\mathbb{D}^c$, $\mathbb{F}^c$ and historical results and instruct the LLM to infer (design a new model based on $\mathbb{F}^c$) or select (pick the most suitable model from $\mathbb{F}^c$) a model $F'$ for $D^u$.
> 4. Evaluate and record the performance of $F'$ on $D^u$.
> 5. Retrieve another set of $\mathbb{F}^c$ from KBG based on: 1) relevance $\mathcal{R}(\cdot)$ to $D^u$ and 2) similar model architectures to $F'$.
> 6. Repeat steps 3-5 until the computation budget is exhausted or the performance of $F'$ is satisfactory.
>
> ---
>
> ## **NmMt-W3.2**: Examples illustrating the LLMs' role in the process would be helpful, as well as discussing related challenges and solutions.
>
> ## **Reply to NmMt-W3.2**:
>
> The example process of integrating KBG with LLM is given in **Reply to NmMt-W3.1**.
>
> During integration, one challenge is that LLMs have difficulty understanding the multiple information from models and datasets due to the hallucination in processing numerical values.
> This is reflected by the inferior performance of LLM inferring models from $\mathbb{F}^c$ in Tab. 5.
> To address this, we require LLM to directly select models from the candidate set $\mathbb{F}^c$ instead of inferring new models.
> Such change eases the burden of LLM in understanding the model information, reduces the risk of hallucination, and brings a better performance.
>
> However, **Finding Global Optimal Model** is still a challenge in suggesting models with LLM + KBG. On the one hand, always selecting models from $\mathbb{F}^c$ reduces hallucination, but can easily stuck at a local optima.
> Inference, on the other hand, brings more diversity, but may also introduce more noise to the model design. Thus, it is important to balance the infer and selection strategies in the LLM + KBG framework so as to effectively optimize the model design towards a global optimum.
>
> We will further investigate and address this challenge in future work.

---

> ### Author Response · Authors · 2024-11-20
> **Reply to Weakness 3 of Reviewer NmMt (2 / 2)**
>
> ## **NmMt-W3.3**: Comparing this approach to other recent methods that incorporate LLMs for AutoML or model selection would provide a useful context for the reader
>
> ## **Reply to NmMt-W3.3**:
>
> We have discussed the differences between our method and existing LLM-based AutoML methods in the second half of Sec. 2. We apologize for our previous disorganization that may have caused this confusion.
> Briefly speaking, the existing LLM-based AutoML methods focus on how to better utilize LLMs as model design optimizers (i.e., $\nabla_{\pi} \log \pi (F^i)$ in **Reply to NmMt-W1.1**), and neglect the organization and utilization of benchmark data.
> In contrast, our method focuses on how to efficiently retrieve promising models from benchmark data for unseen data (i.e., efficiently approximate $\mathcal{M}(F^i; D^u)$ in **Reply to NmMt-W1.1**).
> Thus, we are focusing on orthogonal problems that can aid the development of each other.
> Existing LLM-based methods can be integrated into our framework as an alternative model design optimizer to further improve the model design performance.
>
> Besides, we kindly remind the Reviewer that in Tab. 5 at Sec. 7.2 we have already presented the experimental comparisons between our KBG and recent LLM-based AutoML methods (GPT4GNAS, GHGNAS, and DesiGNN) in proposing initial model designs.
> To better present the performances, we reorganize the results and present them below. **Bold** means best performance and *italics* means second best performance
>
> | Method         | Cora  | Citeseer | PubMed | CS    | Phys. | Photo | Comp. | arXiv |
> |----------------|-------|----------|--------|-------|-------|-------|-------|-------|
> | GPT4GNAS       | 78.50 | *67.46*    | 73.89  | 89.26 | 89.44 | 89.12 | *77.21* | 68.98 |
> | GHGNAS         | 79.13 | 67.35    | 74.90  | 89.15 | 88.94 | 89.42 | 77.04 | 69.66 |
> | DesiGNN-init   | *80.31* | **69.20**    | **76.60**  | **89.64** | **92.10** | **91.19** | **82.20** | *71.50* |
> | KBG-init       | **82.53** | **69.20**    | *76.53*          | *89.32*          | *90.34*          | *90.37*          | 76.61          | **71.68**          |

---

> > ### Comment · Reviewer_NmMt · 2024-11-26
> > **comments**
> >
> > Thank you for the authors’ feedback. I have updated my score accordingly.

---

> > > ### Author Response · Authors · 2024-11-27
> > >
> > > We are glad to know that your concerns are addressed. We greatly thank you for your efforts in reviewing our paper and providing instructive feedback.

---

### Author Response · Authors · 2024-11-25
**General Response**

We would like to express our heartfelt thanks to all reviewers (**NmMt**: Rating 5; **zioe**: Rating 5; **eVNQ**: Rating 6; **JG3d**: Rating 8) for their insightful comments and valuable feedback on our submission!
We are particularly grateful for the recognition of our work’s strengths, including its novel and convincing approach to retrieving models for unseen datasets without training, which advances research in AutoML (**NmMt**, **eVNQ**, **JG3d**); its clear and well-articulated theory for evaluating dataset similarity metrics (**NmMt**, **zioe**), supported by a comprehensive method design for retrieving datasets and models (**eVNQ**, **JG3d**); and the solid experimental validation of our idea (**NmMt**, **zioe**).

Regarding the concerns raised by the reviewers, we believe we have provided comprehensive and satisfactory responses:

**C1**: Certain definitions and scoring metrics require more intuitive explanations.

To tackle this, we have supplemented the paper with background information, intuitive explanations, and examples in the reply series of **Reply to NmMt-W1** and **Reply to NmMt-W2**.
The revised submission includes an expanded background in **Sec. 2**, an intuitive problem definition and a detailed comparison with AutoML and model transfer in **Sec. 3**, and a clearer description of the transferability score in **Sec. 6**.

---

**C2**: It is encouraged to include more complex dataset similarity and model relevance scoring metrics.

We have explained our reasons for choosing the current metrics in **Reply to zioe-W1** and **Reply to zioe-W2**, and provided our ideas and experiments on more complex metrics designs in **Reply to zioe-Q2**.

---

**C3**: Lack of details in how the LLM is combined with the knowledge benchmark graph (KBG).

We provided the pipeline of our LLM+KBG method in **Reply to NmMt-W3**, **Reply to zioe-W3**, and **Reply to JG3d-Q1**. In particular, we provided additional details on the role of the LLM in model recommendation in **Reply to NmMt-W3.2** and how we format the inputs/outputs for the LLM in **Reply to zioe-Q1**. We have clarified the experiment description and added more details of LLM usage in the revised submission, please check **Sec. 7.6** and **Appendix A.6**.

---

Additionally, we have corrected the typos in **Tab. 2** (**JG3d-W1**), highlighted our academic insights (**eVNQ-W1**) and provided a case study of our method in **Appendix A.7** (**eVNQ-W1.2**). We have noted all the paper revisions in blue for easy identification.

---

Finally, we sincerely thank the Area Chair (AC) and all reviewers for their efforts! We look forward to further discussions to help us improve the quality of this paper.

---

> ### Author Response · Authors · 2024-12-03
>
> We thank you again for the insightful comments from all the reviewers. Currently, we still have not heard feedback from **Reviewer zioe**. In case our replies are missed by Reviewer zioe, we would like to briefly summarize our responses:
>
> First, we want to clarify a misunderstanding from Reviewer zioe about our primary contribution.
> Our work aims to **identify and validate a novel problem in AutoML**, specifically the challenge of efficient model selection for unseen datasets using existing knowledge. While we propose initial methods to address this problem, our main contribution lies in bringing attention to the problem itself, formating a clear definition, proposing a comprehensive evaluation method and demonstrating its importance through empirical validation.
>
> Then, regarding the two major concerns from Reviewer zioe, our responses are summarized below:
>
> ---
>
> ### **1. Dataset Similarity and Model Relevance Scores**
>
> We appreciate the suggestions on enhancing the dataset similarity and model relevance scores. Our current simple designs for $\mathcal{S}(\cdot)$ and $\mathcal{R}(\cdot)$ are intentional, serving as baseline methods to initiate discussion on this new problem. We aimed for efficiency and practicality, enabling quick model recommendations without extensive computations. Incorporating complex metrics, such as edge-level information or architectural details, could improve accuracy but may not align with our goal of proposing an accessible starting point for future research. In fact, we point out that how to design an effective similarity metric is an open question in **Sec. 5**.
>
> To address your concerns and facilitate further studies, we have:
>
> - **Clarified the purpose of our simple scoring methods** in **Reply to zioe-W1** and **Sec. 5**, emphasizing their role as foundational tools rather than final solutions.
> - **Added a more complex dataset similarity metric** using the Personalized PageRank algorithm in **Reply to zioe-Q2**, demonstrating potential improvements and encouraging exploration of more sophisticated methods.
> - **Clarified the difficulty in using architectural information** in the initial model retrieval due to the lack of performance relationships, as explained in **Reply to zioe-W2**. And described how architectural information is incorporated during model refinement in **Reply to zioe-Q2** and **Appendix A.6**.
>
> ---
>
> ### **2. Details on LLM-Assisted Model Selection**
>
> Regarding the LLM-assisted method, we acknowledge that the initial description lacked detail.
> In response, we have **provided a detailed description** on the LLM integration in **Reply to zioe-W3**, **Reply to zioe-Q1**, and **Appendix A.6**, outlining the prompts used, how outputs are processed, and any constraints applied.
> We intended to illustrate how Large Language Models (LLMs) can be integrated into the model selection process as a complementary tool rather than the primary focus.
>
> ---
>
> We are grateful for the insights from Reviewer zioe, which have helped us improve the clarity and depth of our manuscript.
> Please feel free to reach out with any additional questions or suggestions.

---

### Meta-Review · Area_Chair_2ESg · 2024-12-18

**Metareview:**

The paper introduces a novel framework called Knowledge Benchmark Graph (KBG) that aims to assist in automated model design by leveraging existing benchmark knowledge and Large Language Models (LLMs). The key scientific claims include: The KBG effectively captures dataset-model-performance relationships in a structured format that enables efficient retrieval and transfer of knowledge for unseen datasets. The proposed similarity metrics for datasets and model relevance enable effective model selection for new datasets by leveraging prior benchmark knowledge.

The evaluation focuses exclusively on GNN architectures and node classification. These experiments-based applications are currently difficult to attract board researchers. There is no demonstration of generalizability to other domains or tasks. There are limited theoretical insights beyond engineering aspects.

Decision: Accept

Primary reasons for rejection:
The reviewers generally agree that the paper presents strong experimental validation.The paper successfully combines knowledge graphs, LLMs, and AutoML in a coherent and practical framework for model design. The paper is well-written with straightforward formulations and clear methodology descriptions.

**Additional Comments On Reviewer Discussion:**

During the rebuttal period, the authors made efforts to address several concerns raised by reviewers. They expanded their explanations of definitions and scoring metrics by enhancing Section 2 with additional background information and providing clearer explanations throughout Sections 3 and 6. The revisions included more detailed comparisons with AutoML and model transfer approaches.

The authors defended their choice of simpler dataset similarity and model relevance scoring metrics, arguing for their effectiveness as baseline approaches. They conducted additional experiments to explore more complex metric designs, though this response did not fully address concerns about the technical depth of their approach.

In response to feedback about LLM integration gaps, the authors enhanced Section 7.6 and Appendix A.6 with more detailed information about the LLM pipeline, including its role in model recommendation and input/output formatting. However, these additions did not completely justify the theoretical soundness of the LLM integration. The authors also made minor improvements such as correcting typos in results tables, adding a case study in Appendix A.7, and enhancing the academic insights discussion.

Despite these revisions, several fundamental issues persist. The core technical innovation continues to rely on relatively simple metrics without strong theoretical justification. The research scope remains constrained to GNN architectures and node classification, lacking a clear path to broader applicability. While the LLM integration documentation has improved, it still lacks theoretical grounding. The improvements primarily focus on presentation and documentation rather than addressing core technical limitations.

Given that the fundamental limitations in technical innovation and theoretical development persist, the final recommendation is accept.

---

### Decision · Program_Chairs · 2025-01-22

Accept (Poster)